# Reflect-then-Plan:
# Offline Model-Based Planning through a *Doubly Bayesian* Lens

Jihwan Jeong [1] [*]   Xiaoyu Wang [1]   Jingmin Wang [1]   Scott Sanner [1]   Pascal Poupart [2]

## Abstract

Offline reinforcement learning (RL) is crucial when online exploration is costly or unsafe but often struggles with high epistemic uncertainty due to limited data. Most of existing methods rely on fixed conservative policies, restricting adaptivity and generalization. To address this, we propose Reflect-then-Plan (RefPlan), a novel *doubly Bayesian* offline model-based (MB) planning approach. RefPlan unifies uncertainty modeling and MB planning by recasting planning as Bayesian posterior estimation. At deployment, it updates a belief over environment dynamics using real-time observations, incorporating uncertainty into MB planning via marginalization. Empirical results on standard benchmarks show that RefPlan significantly improves the performance of conservative offline RL policies. In particular, RefPlan maintains robust performance under high epistemic uncertainty and limited data, while demonstrating resilience to changing environment dynamics, improving the flexibility, generalizability, and robustness of offline-learned policies.

## 1. Introduction

Recent advances in offline reinforcement learning (RL) enable learning performant policies from static datasets (Levine et al., 2020; Kumar et al., 2020), making it appealing when online exploration is costly or unsafe.

The agent's inability to gather more experiences have severe implications. In particular, it becomes practically impossible to precisely identify the true Markov decision process (MDP) with a limited dataset, as it only covers a portion of the entire state-action space, leading to high *epistemic uncertainty* for states and actions outside the data distribution.

Most offline RL methods aim to learn a conservative policy that stays close to the data distribution, thus steering away from high epistemic uncertainty.

While incorporating conservatism into offline learning has proven effective (Jin et al., 2021; Kumar et al., 2020), it can result in overly restrictive policies that lack generalizability. Most methods learn a Markovian policy that relies solely on the current state, leading the agent to potentially take poor actions in unexpected states during evaluation. Model-based (MB) planning can enhance the agent's responsiveness during evaluation (Sikchi et al., 2021; Argenson & Dulac-Arnold, 2021; Zhan et al., 2022), but it still primarily addresses epistemic uncertainty through conservatism.

Noting this challenge, Chen et al. (2021) and Ghosh et al. (2022) propose to learn an *adaptive* policy that can reason about the environment and accordingly react at evaluation. Essentially, they formulate the offline RL problem as a partially observable MDP (POMDP)—where the partial observability relates to the agent's epistemic uncertainty, aka *Epistemic POMDP* (Ghosh et al., 2021). Thus, learning an adaptive policy involves approximately inferring the *belief state* from the history of transitions experienced by the agent and allowing the policy to condition on this belief state.

While learning an adaptive policy can help make the agent more flexible and generalizable, it still heavily depends on the training phase. Our empirical evaluation demonstrates that a learned policy—whether it be adaptive or fixed—can be significantly strengthened by incorporating MB planning. However, existing MB planning methods fall short in adequately addressing the agent's epistemic uncertainty, and it remains elusive how one can effectively incorporate the uncertainty into planning.

We propose **Reflect-then-Plan (RefPlan)**, a novel *doubly Bayesian* approach for offline MB planning. RefPlan combines epistemic uncertainty modeling with MB planning in a unified probabilistic framework, inspired by the control-as-inference paradigm (Levine, 2018). RefPlan adapts Bayes-adaptive deep RL techniques (Zintgraf et al., 2020; Dorfman et al., 2021) to infer a posterior belief distribution from past experiences during test time (*Reflect*). To harness this uncertainty for planning, we recast planning as Bayesian posterior

---

[*]Currently at Google Research, Mountain View, CA. [1]University of Torornto, Toronto [2]University of Waterloo, Waterloo. Correspondence to: Jihwan Jeong <jihwanjeong@google.com>.

*Proceedings of the $42^{nd}$ International Conference on Machine Learning*, Vancouver, Canada. PMLR 267, 2025. Copyright 2025 by the author(s).

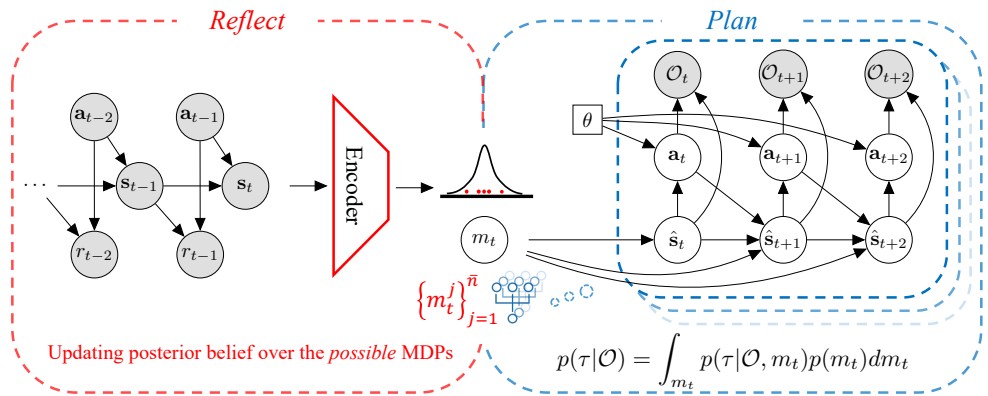

Figure 1: Schematic illustration of RefPlan. (*Reflect*) At time $t$, real-time experiences $\tau_{:t} = (\mathbf{s}_0, \mathbf{a}_0, r_0, \dots, \mathbf{s}_t)$ are used to infer the posterior belief $m_t$ over environments using a variational autoencoder. Unlike prior methods, RefPlan learns diverse dynamics models conditioned on $m_t$, capturing different transition and reward functions. (*Plan*) Offline planning is framed as probabilistic inference, where the posterior over optimal plans $p(\tau|\mathcal{O})$ ($\mathcal{O}$ denotes optimality variables in the control-as-inference framework) is inferred. Importantly, a prior $p(\tau)$ is incorporated by learning $\pi_\theta$ via offline policy learning. By marginalizing out $m_t$, RefPlan addresses epistemic uncertainty, enhancing $\pi_\theta$ for better adaptivity and generalizability.

estimation (*Plan*). By marginalizing over the agent's epistemic uncertainty, RefPlan effectively considers a range of possible scenarios beyond the agent's immediate knowledge, resulting in a posterior distribution over optimized plans under the learned model (Figure 1).

In our experiments, we demonstrate that RefPlan can be integrated with various offline RL policy learning algorithms to consistently boost their test-time performance in standard offline RL benchmark domains (Fu et al., 2020). RefPlan maintains robust performance under high epistemic uncertainty, demonstrating superior resilience when faced with out-of-distribution states, shifts in environment dynamics, or limited data availability, consistently outperforming competing methods in these challenging scenarios.

## 2. Preliminaries

We study RL in the framework of *Markov decision processes* (MDPs), characterized by a tuple $\mathcal{M} = (\mathcal{S}, \mathcal{A}, T, r, d_0, \gamma)$. The state ($\mathcal{S}$) and action ($\mathcal{A}$) spaces are continuous, $T(\mathbf{s}'|\mathbf{s}, \mathbf{a})$ is the transition probability, $r(\mathbf{s}, \mathbf{a})$ is the reward function, $d_0$ is the initial state distribution, and $\gamma \in [0, 1]$ is the discount factor. The *model* of the environment refers to the transition and reward functions. The goal of RL is to find an optimal policy $\pi^*$ which maximizes the expected discounted return, $\mathbb{E}_{\mathbf{s}_0 \sim d_0, \mathbf{s}_t \sim T, \mathbf{a}_t \sim \pi^*}[\sum_{t=0}^{\infty} \gamma^t r(\mathbf{s}_t, \mathbf{a}_t)]$.

**Offline MB planning** Given an offline dataset $\mathcal{D} = \{(\mathbf{s}_i, \mathbf{a}_i, r_i, \mathbf{s}'_i)\}_{i=1}^{N}$ collected by a behavior policy $\beta$, model-based (MB) methods train a predictive model $\hat{p}_\psi(\mathbf{s}', r|\mathbf{s}, \mathbf{a})$ via maximum likelihood estimation (MLE), minimizing

$L(\psi) = \mathbb{E}_{\mathcal{D}}[-\log \hat{p}_\psi(\mathbf{s}', r|\mathbf{s}, \mathbf{a})]$. The learned model $\hat{p}_\psi$ then generates imaginary data $\mathcal{D}_{\text{model}}$ to aid offline policy learning.

Here, we focus on using learned models for *test-time planning*. MB planning typically employs model predictive control (MPC), where at each step, a trajectory optimization (TrajOpt) method *re-plans* to maximize the expected $H$-step return under $\hat{p}_\psi$, incorporating a value function $V_\phi$ for long-term rewards (Lowrey et al., 2018):

$$\mathbf{a}^*_{t:t+H} = \arg\max_{\mathbf{a}_{t:t+H}} \mathbb{E}_{\hat{p}_\psi}[R_H(\mathbf{s}_t, \mathbf{a}_{t:t+H})], \quad (1)$$

where $R_H(\mathbf{s}_t, \mathbf{a}_{t:t+H}) := \sum_{h=0}^{H-1} \gamma^h \hat{r}_\psi(\hat{\mathbf{s}}_{t+h}, \mathbf{a}_{t+h}) + \gamma^H V_\phi(\hat{\mathbf{s}}_{t+H})$ is the return of a candidate sequence $\mathbf{a}_{t:t+H}$ under $\hat{p}_\psi$, with $\hat{r}_\psi$ denoting the learned reward model.

MPPI is a TrajOpt method that optimizes trajectories by sampling $\bar{N}$ action sequences, weighting them via a softmax function with inverse temperature $\kappa$ based on returns (Nagabandi et al., 2019). The optimized action is $\mathbf{a}^*_{t+h} = \frac{\sum_{n=1}^{\bar{N}} \exp(\kappa R_H^n) \cdot \mathbf{a}_{t+h}^n}{\sum_{n=1}^{\bar{N}} \exp(\kappa R_H^n)}$ where $R_H^n$ is the return of the $n$th trajectory. MBOP (Argenson & Dulac-Arnold, 2021) adapts MPPI for offline settings by sampling actions from a behavior-cloning (BC) policy with smoothing.

**The control-as-inference framework** The control-as-inference framework reformulates RL as a probabilistic inference problem (Levine, 2018), introducing auxiliary binary variables $\mathcal{O}_t$, where $\mathcal{O}_t = 1$ denotes that $(\mathbf{s}_t, \mathbf{a}_t)$ is optimal. The likelihood of optimality for a trajectory $\tau_{t:t+H} = (\mathbf{s}_t, \mathbf{a}_t, \dots, \mathbf{s}_{t+H})$ is:

**Definition 1** (The optimality likelihood). *For $\tau_{t:t+H}$, let $\mathcal{O} = 1$ if $\mathcal{O}_{t+h} = 1 \; \forall h$. Then,*

$$p(\mathcal{O} = 1|\tau) \propto \prod_h p(\mathcal{O}_{t+h} = 1|\mathbf{s}_{t+h}, \mathbf{a}_{t+h}). \quad (2)$$

Assuming $p(\mathcal{O}_{t+h} = 1|\mathbf{s}_{t+h}, \mathbf{a}_{t+h}) \propto \exp(\kappa \cdot r_{t+h})$, we obtain $p(\mathcal{O}|\tau) \propto \exp(\kappa \cdot \sum_h r_h)$.[1] Thus, expected return maximization reduces to posterior inference over trajectories given that all time steps are optimal under the probabilistic graphical model (PGM) (see Figure 2, left):

$$p(\tau|\mathcal{O}) \propto p(\tau, \mathcal{O}) =$$
$$p(\mathbf{s}_t) \prod_{h=1}^{H} p(\mathcal{O}_{t+h}|\mathbf{s}_{t+h}, \mathbf{a}_{t+h}) p(\mathbf{s}_{t+h+1}|\mathbf{s}_{t+h}, \mathbf{a}_{t+h}). \quad (3)$$

Typically, the prior over actions is assumed to be uniform or implicitly defined by rewards (Levine, 2018; Piché et al., 2019). However, in Section 3.1, we explicitly model the action prior to formalize an offline MB planning framework, enabling policy refinement via MB planning.

**Epistemic POMDP, BAMDP, and Offline RL**  A partially observable MDP (POMDP) extends MDPs to scenarios with incomplete state information. POMDPs can be reformulated as belief-state MDPs, where a *belief*—a probability distribution over states—represents the uncertainty over states given the agent's prior observations and actions (Kaelbling et al., 1998).

Unlike an ordinary POMDP, *epistemic POMDPs* address generalization to unseen test conditions in RL (Ghosh et al., 2021). In this scenario, the agent experiences partial observability entirely due to its epistemic uncertainty about the identity of the true environment $\mathcal{M}$ at test time. Specifically, at test time, the agent's goal is to maximize the expected return $\mathbb{E}_{\mathcal{M} \sim p(\mathcal{M}|\mathcal{D})}[\sum_t \gamma^t r_t]$ under the posterior $p(\mathcal{M}|\mathcal{D})$ given the train data $\mathcal{D}$. Thus, an epistemic POMDP is an instance of Bayes-adaptive MDP (BAMDP) (Duff, 2002; Kaelbling et al., 1998) or Bayesian RL (Ghavamzadeh et al., 2015). Compared to a standard BAMDP, epistemic POMDP puts focus on the agent's test time evaluation performance. For a thorough definition of BAMDPs, please check Appendix A.1.

In this work, we view offline RL as epistemic POMDP, following Ghosh et al. (2022), drawing connections to Bayesian approaches. That is, limited coverage of the state-action space in the offline dataset induces epistemic uncertainty about dynamics beyond the data distribution. Failure to manage this uncertainty can result in catastrophic outcomes, particularly when an offline-trained agent encounters

unseen states or slightly altered dynamics during deployment, leading to arbitrarily poor performance.

To address these challenges, we provide a Bayesian take on the offline RL problem, enabling reasoning over the agent's uncertainty through a prior belief $b_0 = p(\mathcal{M})$, updated to a posterior $b_t = p(\mathcal{M}|\tau_{:t})$ as new experiences $\tau_{:t}$ are gathered during deployment. However, computing the exact posterior belief is generally intractable. Therefore, in Section 3, we tackle this challenge by approximating the belief distribution through variational inference techniques adapted from meta-RL approaches (Zintgraf et al., 2020; Dorfman et al., 2021).

## 3. RefPlan: a Probabilistic Framework for Offline Planning

In this section, we seek to address the following question:

*How can we leverage a learned model at test time for **enhancing an offline-trained agent**?*

To tackle this, we introduce *RefPlan*, a novel probabilistic framework for offline MB planning that allows an agent to reason with its uncertainty during deployment. Our approach is developed in three parts.

First, in Section 3.1, we derive a sampling-based offline MB planning algorithm from the control-as-inference perspective. This view is crucial as it provides a principled foundation for our framework and formally justifies using an offline-trained policy as a prior to guide planning. Next, in Section 3.2, we detail our approach for modeling the agent's epistemic uncertainty over the environment dynamics by adapting recent variational inference techniques from meta-RL (Zintgraf et al., 2020; Dorfman et al., 2021).

While these individual components are built upon concepts from prior work, our primary conceptual novelty lies in their synthesis. In Section 3.3, we unify the probabilistic planning formulation with the learned epistemic uncertainty. This integration allows the agent to plan under the learned models and adapt in real-time while accounting for its uncertainty.

### 3.1. Offline Model-Based Planning as Probabilistic Inference

We recast offline MB planning within the *control-as-inference* framework, treating planning as a posterior inference problem. This approach enables the agent to optimize its actions by reasoning over the learned dynamics model and prior knowledge obtained from offline training. Central to this formulation is the use of a *prior policy*, which guides the agent's plans based on knowledge learned offline.

We start by formalizing the concept of a prior policy, which lay the basis for our Bayesian formulation of the offline MB

---

[1]While we use an exponential form for $p(\mathcal{O}|\tau)$, other monotonic functions are possible (Okada & Taniguchi, 2020). We simplify notation by writing $\mathcal{O}_t = 1$ as $\mathcal{O}_t$ and $\tau_{t:t+H}$ as $\tau$.

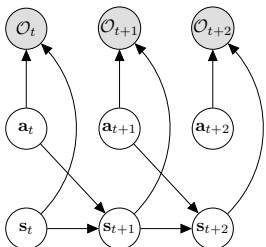 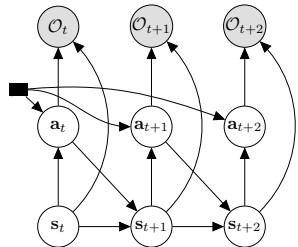 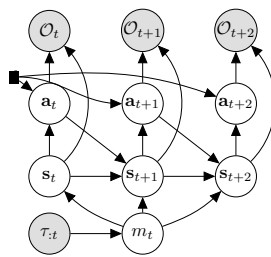

Figure 2: PGMs for the control-as-inference framework, offline MB planning, and RefPlan. **(Left)** States evolve within the learned model, with actions and states influencing optimality. Optimality variables act like observations in a hidden Markov model, framing planning as inferring the posterior over actions given optimality. **(Middle)** In offline MB planning, actions follow the prior policy $\pi_{\mathrm{p}}$: $\mathbf{a}_t \sim \pi_{\mathrm{p}}(\cdot|\mathbf{s}_t; \theta)$. **(Right)** RefPlan uses past experiences $\tau_{:t}$ to infer $m_t$, the agent's belief about the environment, and computes the expected optimal action sequence by marginalizing over $m_t$.

planning process.

**Definition 2** (Prior policy). *A prior policy* $\pi_{\mathrm{p}} : \mathcal{S} \rightarrow \mathcal{P}(\mathcal{A})$ *is a policy learned from an offline RL algorithm* $\mathfrak{L}$ *using the dataset* $\mathcal{D}$.

The prior policy, parameterized by $\theta$, is provided by an offline learning algorithm $\mathfrak{L}$, such as CQL (Kumar et al., 2020) or BC, and must be considered by the offline MB planner when optimizing the planning objective in (1).

In the offline setting, we aim to enhance the prior policy $\pi_{\mathrm{p}}$ via MB planning at test time by inferring the posterior over $\mathbf{a}_{t:t+H}$, conditioned on the optimality observations $\mathcal{O}_{t+h}$ predicted by the learned model $\hat{p}_{\psi}$. At time $t$, we seek to compute $p(\mathbf{a}_{t:t+H}|\mathcal{O})$, as shown in Figure 2 (middle).

*The key distinction in this setup from the original control-as-inference framework is the inclusion of the prior policy, which serves as a source for action sampling during planning.* Given $\pi_{\mathrm{p}}$ and $\hat{p}_{\psi}$, we can define the prior distribution over the trajectory $\tau$ as follows:

$$p(\tau) = \prod_{h=0}^{H-1} \pi_{\mathrm{p}}(\mathbf{a}_{t+h}|\mathbf{s}_{t+h})\hat{p}_{\psi}(\mathbf{s}_{t+h+1}|\mathbf{s}_{t+h}, \mathbf{a}_{t+h}). \quad (4)$$

Sampling trajectories from this prior, $p(\tau)$, is straightforward through forward sampling, where actions are drawn from $\pi_{\mathrm{p}}$ and state transitions are generated using $\hat{p}_{\psi}$.

Computing the exact posterior $p(\tau|\mathcal{O})$ is intractable due to the difficulty of calculating the marginal $p(\mathcal{O})$. However, importance sampling offers a practical method to estimate the posterior expectation over $\mathbf{a}_{t:t+H}$. To demonstrate, we first expand the posterior using Bayes' rule:

$$p(\tau|\mathcal{O}) \propto p(\mathcal{O}|\tau)p(\tau) \propto \exp\left(\kappa \sum_{h=0}^{H-1} r_{t+h}\right)$$

$$\left[\prod_{h=0}^{H-1} \hat{p}_{\psi}(\mathbf{s}_{t+h+1}|\mathbf{s}_{t+h}, \mathbf{a}_{t+h})\pi_{\mathrm{p}}(\mathbf{a}_{t+h}|\mathbf{s}_{t+h})\right].$$

Then, we can estimate the expected value of an arbitrary function $f(\mathbf{a}_{t:t+H})$ under $p(\tau|\mathcal{O})$. That is,

$$\mathbb{E}_{p(\tau|\mathcal{O})}[f(\mathbf{a}_{t:t+H})] = \int_{\tau} f(\mathbf{a}_{t:t+H}) \, p(\tau|\mathcal{O}) \, d\tau$$

$$= \int_{\tau} f(\mathbf{a}_{t:t+H}) \frac{\alpha \cdot \exp\left(\kappa \sum_h r_{t+h}\right)}{p(\mathcal{O})} \, p(\tau) \, d\tau$$

$$= \frac{\mathbb{E}_{p(\tau)}\left[f(\mathbf{a}_{t:t+H}) \, \exp\left(\kappa \sum_h r_{t+h}\right)\right]}{\mathbb{E}_{p(\tau)}\left[\exp\left(\kappa \sum_h r_{t+h}\right)\right]}. \quad (5)$$

In the last step, we used $p(\mathcal{O}) = \int_{\tau} p(\mathcal{O}|\tau)p(\tau)d\tau = \alpha \, \mathbb{E}_{p(\tau)}\left[\exp\left(\kappa \sum_h r_{t+h}\right)\right]$ and the proportionality coefficient $\alpha > 0$ cancels out.

Thus, the posterior expectation over $\mathbf{a}_{t:t+H}$ can be obtained with $f(\mathbf{a}_{t:t+H}) = \mathbf{a}_{t:t+H}$ as below.

$$\mathbb{E}_{p(\tau|\mathcal{O})}[\mathbf{a}_{t:t+H}] = \frac{\mathbb{E}_{p(\tau)}\left[\mathbf{a}_{t:t+H} \, \exp\left(\kappa \sum_h r_{t+h}\right)\right]}{\mathbb{E}_{p(\tau)}\left[\exp\left(\kappa \sum_h r_{t+h}\right)\right]} \quad (6)$$

$$\approx \sum_{n=1}^{\bar{N}} \left(\frac{\exp\left(\kappa \sum_h r_{t+h}^n\right)}{\sum_{i=1}^{\bar{N}} \exp\left(\kappa \sum_h r_{t+h}^i\right)}\right) \mathbf{a}_{t:t+H}^n. \quad (7)$$

That is, we estimate the posterior mean by sampling $\bar{N}$ trajectories from $p(\tau)$ with $\pi_{\mathrm{p}}$ and $\hat{p}_{\psi}$, then computing the weighted sum of the actions. Each weight $w^n := \frac{\exp(\kappa \sum_h r_{t+h})}{\sum_{i=1}^{\bar{N}} \exp(\kappa \sum_h r_{t+h}^i)}$ is proportional to the exponentiated MB return of the $n$th trajectory, assigning higher weights to plans with better returns. This helps the agent select actions likely to improve on those from the prior policy.

We note that (7) can also be derived from an optimization perspective. Specifically, LOOP (Sikchi et al., 2021) constrains the distribution over plans by minimizing the KL divergence from the prior policy. In LOOP, the variance of values generated by the model ensemble is penalized to mitigate uncertainty; however, the agent's epistemic uncertainty is not explicitly modeled and fully addressed. By

contrast, by viewing offline RL as an epistemic POMDP and formulating it as a probabilistic inference problem, we can directly incorporate the agent's epistemic uncertainty into MB planning by approximately learning the belief distribution, which we delve into in the next part.

### 3.2. Learning Epistemic Uncertainty via Variational Inference

Although offline RL can be framed as a BAMDP, obtaining an exact posterior belief update is impractical. Inspired by Zintgraf et al. (2020) and Dorfman et al. (2021), we introduce a latent variable $m$ to approximate the underlying MDP. We assume that knowing the posterior distribution $p(m|\tau_{:t})$ is sufficient for planning under epistemic uncertainty. As a result, transitions and rewards are assumed to depend on $m$, i.e., $T(\mathbf{s}_{t+1}|\mathbf{s}_t, \mathbf{a}_t, m)$ and $r(\mathbf{s}_t, \mathbf{a}_t, m)$. When $p(m|\tau_{:t})$ is accurate and $\tau_{:t}$ is in-distribution, $T$ and $r$ will closely match the transitions in $\mathcal{D}$. For out-of-distribution (OOD) $\tau_{:t}$, the posterior over $m$ allows modeling diverse scenarios for $T$ and $r$.

Given a trajectory $\tau_{:t}$, consider the task of maximizing its likelihood, conditioned on the actions. Conditioning on the actions is essential because they are generated by a policy—$\beta$ during training and $\pi_{\mathrm{p}}$ at evaluation—and are not modeled by the environment. Although directly optimizing the likelihood $p(\mathbf{s}_0, r_0, \mathbf{s}_1, r_1, \ldots, \mathbf{s}_{t+1}|\mathbf{a}_0, \ldots, \mathbf{a}_t)$ is intractable, we can maximize the ELBO as in VariBAD by introducing an encoder $q_\varphi$ and a decoder $\hat{p}_\psi$:

$$\log p(\mathbf{s}_0, r_0, \ldots, \mathbf{s}_{t+1}|\mathbf{a}_0, \ldots, \mathbf{a}_t)$$

$$= \log \int_{m_t} p(\mathbf{s}_0, r_0, \ldots, \mathbf{s}_{t+1}, m_t \mid \mathbf{a}_0, \ldots, \mathbf{a}_t)\, dm_t$$

$$= \log \mathbb{E}_{m_t \sim q_\varphi(\cdot|\tau_{:t})} \left[ \frac{p(\mathbf{s}_0, r_0, \ldots, \mathbf{s}_{t+1}, m_t \mid \mathbf{a}_0, \ldots, \mathbf{a}_t)}{q_\varphi(m_t|\tau_{:t})} \right]$$

$$\geq \mathbb{E}_{m_t \sim q_\varphi(\cdot|\tau_{:t})}[\log \hat{p}_\psi(\mathbf{s}_0, \ldots, \mathbf{s}_{t+1}|m_t, \mathbf{a}_0, \ldots, \mathbf{a}_t)]$$

$$- KL(q_\varphi(m_t|\tau_{:t})||p(m_t)) = ELBO_t(\varphi, \psi). \quad (8)$$

The encoder $q_\varphi$ is parameterized as an RNN followed by a fully connected layer that outputs Gaussian parameters $\mu(\tau_{:t})$ and $\log \sigma^2(\tau_{:t})$. Thus, $m_t \sim q_\varphi(\cdot|\tau_{:t}) = \mathcal{N}\left(\mu(\tau_{:t}), \sigma^2(\tau_{:t})\right)$. The KL term regularizes the posterior with the prior $p(m_t)$, which is a standard normal at $t = 0$ and the previous posterior $q_\varphi(\cdot|\tau_{:t-1})$ for subsequent time steps. The decoder $\hat{p}_\psi$ learns the transition dynamics and reward function of the true MDP. This becomes clear when we observe that the first term in $ELBO_t$ corresponds to the reconstruction loss, which can be decomposed as follows:

$$\log \hat{p}_\psi(\mathbf{s}_0, r_0, \ldots, \mathbf{s}_{t+1}|m_t, \mathbf{a}_0, \ldots, \mathbf{a}_t) \quad (9)$$

$$= \log p(\mathbf{s}_0|m_t) + \sum_{h=0}^{t} \big[ \log \hat{p}_\psi(\mathbf{s}_{h+1}|\mathbf{s}_h, \mathbf{a}_h, m_t)$$

$$+ \log \hat{p}_\psi(r_{h+1}|\mathbf{s}_h, \mathbf{a}_h, m_t) \big].$$

Here, $\hat{p}_\psi$ learns to predict future states and rewards conditioned on the latent variable $m_t$. The encoder captures the agent's epistemic uncertainty, while the decoder provides predictions about the environment under different latent variables $m_t$. To sum up, we train a variational autoencoder (VAE) via $\max_{\phi,\psi}\ \mathbb{E}_\mathcal{D}\left[\sum_{t=0}^{T} ELBO_t(\phi, \psi)\right]$ using trajectories sampled from the offline dataset $\mathcal{D}$.

*Unlike VariBAD, where the decoder is only used to train the encoder, we also use $\hat{p}_\psi$ for MB planning.* To improve $\hat{p}_\psi$'s accuracy, we employ a two-stage training procedure: first, the VAE is trained with ELBO; then, the encoder is frozen and $\hat{p}_\psi$ is further finetuned using the MLE objective:

$$L(\psi) = \mathbb{E}_\tau \left[ \sum_{h=0}^{H-1} \mathbb{E}_{m_h}[-\log \hat{p}_\psi(\mathbf{s}_{h+1}, r_h|\mathbf{s}_h, \mathbf{a}_h, m_h)] \right].$$

Trajectory segments of length $H$ are sampled from $\mathcal{D}$. At each step $h \in [0, H)$, the encoder $q_\varphi(\cdot|\tau_{:h})$ samples $m_h$, enabling computation of the inner expectation and refining $\hat{p}_\psi$ for improved predictions.

### 3.3. Integrating Epistemic Uncertainty into Model-Based Planning

Building on the probabilistic inference formulation of offline MB planning and the representation of epistemic uncertainty via variational inference in the BAMDP framework, we introduce RefPlan. This offline MB planning algorithm integrates epistemic uncertainty into the planning process, improving decision-making and enhancing the performance of any offline-learned prior policy during test time.

Assume we have a sample $m_t \sim q_\varphi(m|\tau_{:t})$, representing the agent's belief about the environment at time $t$. Our goal is to use this posterior to enhance test-time planning. In Section 3.1, we have computed $p(\tau|\mathcal{O})$ using the learned models $\hat{p}_\psi$ and the prior policy $\pi_{\mathrm{p}}$. By introducing the latent variable $m$ to capture epistemic uncertainty, we extend the transition and reward functions to depend on $m$, giving the dynamics $\hat{p}_\psi(\mathbf{s}_{t+1}|\mathbf{s}_t, \mathbf{a}_t, m_t)$ and rewards $r(\mathbf{s}_t, \mathbf{a}_t, m_t)$, resulting in the following conditional trajectory distribution:

$$p(\tau|\mathcal{O}, m_t) \propto p(\mathcal{O}|\tau, m_t)p(\tau|m_t)$$

$$\propto \exp\left( \kappa \sum_{h=0}^{H-1} r(\mathbf{s}_{t+h}, \mathbf{a}_{t+h}, m_t) \right)$$

$$\times \left[ \prod_{h=0}^{H-1} \hat{p}_\psi(\mathbf{s}_{t+h+1}|\mathbf{s}_{t+h}, \mathbf{a}_{t+h}, m_t)\pi_{\mathrm{p}}(\mathbf{a}_{t+h}|\mathbf{s}_{t+h}) \right].$$

Thus, we can apply the sampling-based posterior estimation in (7) to compute $\mathbb{E}_{p(\tau|\mathcal{O},m_t)}[\mathbf{a}_{t:t+H}]$.

A practical approach to handle epistemic uncertainty is to marginalize over the latent variable $m_t$, effectively averag-

Table 1: Normalized score performance when trained on ME and tested starting from OOD states sampled from R. For each prior policy, we report the performance of the prior policy (Prior), performance with LOOP, and performance with RefPlan. The 'Orig' column shows the performance of the original prior policy on the ME dataset for reference. Best results under OOD conditions are in **bold**.

| Prior Policy | Environment | Orig | Prior | + LOOP | + RefPlan |
|---|---|---|---|---|---|
| CQL | Hopper | 111.4 | 84.21 | **90.01** | 89.39 |
| | HalfCheetah | 98.3 | 71.81 | 85.48 | **85.92** |
| | Walker2d | 108.9 | 65.01 | 85.95 | **92.61** |
| MAPLE | Hopper | 46.9 | 39.60 | **45.78** | 42.50 |
| | HalfCheetah | 64.0 | 62.31 | 91.41 | **91.84** |
| | Walker2d | 111.8 | 66.82 | 74.05 | **87.82** |
| COMBO | Hopper | 105.6 | 81.75 | 80.67 | **85.51** |
| | HalfCheetah | 97.6 | 57.89 | 79.57 | **84.49** |
| | Walker2d | 108.3 | 62.54 | 66.88 | **72.82** |

ing over possible scenarios. This results in the marginal posterior $p(\tau|\mathcal{O})$. Although directly computing this marginal posterior is challenging, we can estimate the expectation of optimal plans using the law of total expectation:

$$\mathbb{E}_{p(\tau|\mathcal{O})}[\mathbf{a}_{t:t+H}] = \mathbb{E}_{m_t \sim q_\varphi(\cdot|\tau_{:t})}\left[\mathbb{E}_{p(\tau|\mathcal{O},m_t)}[\mathbf{a}_{t:t+H} \mid m_t]\right].$$

The inner expectation follows (7), with states and rewards sampled from $\hat{p}_\psi$, conditional on $m_t$. The outer expectation over $m_t$ is computed using Monte Carlo sampling with $\bar{n}$ samples, giving us:

$$\mathbb{E}_{p(\tau|\mathcal{O})}[\mathbf{a}_{t:t+H}] \approx$$
$$\frac{1}{\bar{n}}\sum_{j=1}^{\bar{n}}\left[\sum_{n=1}^{\bar{N}}\left(\frac{\exp\left(\kappa\sum_h r_{t+h}^{n,j}\right)}{\sum_{i=1}^{\bar{N}}\exp\left(\kappa\sum_h r_{t+h}^{i,j}\right)}\right)\mathbf{a}_{t:t+H}^n\right], \quad (10)$$

where $r_{t+h}^{n,j} = r(\mathbf{s}_{t+h}^{n,j}, \mathbf{a}_{t+h}^n, m_t^j)$ and $\mathbf{s}_{t+h+1}^{n,j} \sim \hat{p}_\psi(\cdot|\mathbf{s}_{t+h}^{n,j}, \mathbf{a}_{t+h}^n, m_t^j)$. Figure 2 (right) illustrates how Ref-Plan leverages the agent's past experiences $\tau_{:t}$ to shape epistemic uncertainty through the latent variable $m_t$ and enhances the prior policy $\pi_p$ through posterior inference. Algorithm 2 in the appendix summarizes RefPlan.[2] Additionally, following Sikchi et al. (2021), we apply an uncertainty penalty based on the variance of the returns predicted by the learned model ensemble.

## 4. Experiments

In this part, we answer the following research questions: **(RQ1)** How does RefPlan perform when the agent is initialized in a way that induces high epistemic uncertainty due to OOD states? **(RQ2)** Can RefPlan effectively improve policies learned from diverse offline policy learning algorithms?

**(RQ3)** How does RefPlan perform when trained on limited offline datasets that increase epistemic uncertainty by restricting the datasets' coverage of the state-action space? **(RQ4)** How robust is RefPlan when faced with shifts in environment dynamics at test time?

We evaluate these RQs using the D4RL benchmark (Fu et al., 2020) and its variations, focusing on locomotion tasks in *HalfCheetah*, *Hopper*, and *Walker2d* environments, each with five configurations: *random* (R), *medium* (M), *medium-replay* (MR), *medium-expert* (ME), and *full-replay* (FR).

**Metrics:** For RQ1-RQ3, we compare normalized scores averaged over 3 seeds, with 100 for online SAC and 0 for a random policy, scaled linearly in between, using the D4RL library (Fu et al., 2020). For RQ4, we report average returns.

**Baselines:** RefPlan is designed to improve any offline learned policy through planning. We have obtained prior policies using model-free methods (CQL, EDAC) and MB methods (MOPO, COMBO, MAPLE). Among offline MB planning methods, we use LOOP, which is designed to enhance prior policies and outperforms methods like MBOP. Therefore, for each prior policy, we compare its original performance to its performance when augmented with LOOP or RefPlan for test-time planning.

### 4.1. Epistemic Uncertainty from OOD States

To address RQ1, we assessed RefPlan's robustness under high epistemic uncertainty caused by OOD initialization. Prior policies were trained on the ME dataset and evaluated on the states from the R dataset. We tested three prior policies: CQL, MAPLE, and COMBO (Table 1).

Across all environments, RefPlan consistently mitigated performance degradation due to OOD initialization, with particularly notable improvements in *HalfCheetah* and *Walker2d*. For instance, when MAPLE was used as the prior policy in

---

[2]Direct planning with sampling methods like SIR (Skare et al., 2003) may be better for multi-modal problems. However, our approach using (10) yields strong empirical results, so we leave direct sampling for future work.

Table 2: Normalized scores of offline RL algorithms on D4RL MuJoCo Gym environments (3 seeds). For each prior policy, we show its original performance and its performance augmented with LOOP or RefPlan (Ours) for MB planning during testing. **Bold** indicates the best performance, while underline denotes cases where confidence intervals significantly overlap between two methods.

| | | CQL | | | EDAC | | | MOPO | | | COMBO | | | MAPLE | | |
|---|---|---|---|---|---|---|---|---|---|---|---|---|---|---|---|---|
| | | Orig | LOOP | Ours | Orig | LOOP | Ours | Orig | LOOP | Ours | Orig | LOOP | Ours | Orig | LOOP | Ours |
| Hopper | R | 1.0 | 1.1 | 1.2 | 23.6 | 23.5 | 23.5 | 32.2 | 32.4 | 32.4 | 6.3 | 6.2 | 6.0 | 31.5 | 31.8 | 31.6 |
| | M | 66.9 | 73.9 | **85.1** | 101.5 | 101.5 | 101.5 | 66.9 | 67.5 | 67.7 | 60.9 | 67.9 | **77.2** | 29.4 | **33.7** | 32.8 |
| | MR | 94.6 | 97.5 | **98.1** | 100.4 | 101.0 | 101.1 | 90.3 | 93.6 | **94.5** | 101.1 | 101.4 | 101.8 | 61.0 | 77.7 | **82.6** |
| | ME | 111.4 | 111.6 | **112.1** | 106.7 | 104.7 | **109.9** | 91.3 | 82.7 | **96.5** | 105.6 | 78.4 | **107.8** | 46.9 | 53.4 | **57.8** |
| | FR | 104.2 | 106.2 | **107.6** | 106.6 | 107.0 | 107.2 | 73.2 | 55.6 | **77.2** | 89.9 | 54.9 | 84.1 | 79.1 | 77.0 | **91.7** |
| HalfCheetah | R | 19.9 | 21.4 | 21.2 | 22.5 | 25.8 | 25.9 | 29.8 | 31.5 | **33.0** | 40.3 | 40.0 | 40.7 | 33.5 | 34.9 | 35.0 |
| | M | 47.4 | **57.1** | 56.5 | 63.8 | **73.0** | 71.4 | 42.8 | 58.4 | **59.8** | 67.2 | 73.2 | **77.4** | 68.8 | 72.9 | **74.6** |
| | MR | 47.0 | 52.1 | **54.1** | 61.8 | 66.9 | 66.5 | 70.6 | 71.8 | **73.8** | 73.0 | 71.2 | **75.0** | 71.5 | 74.7 | **76.3** |
| | ME | 98.3 | 104.0 | **108.5** | 100.8 | 107.1 | **108.8** | 73.5 | 94.5 | **96.6** | 97.6 | 110.3 | 110.3 | 64.0 | 91.9 | **92.8** |
| | FR | 77.5 | 81.8 | **86.7** | 81.7 | 87.5 | **88.5** | 81.7 | 88.2 | **90.8** | 71.8 | 82.6 | **86.3** | 66.8 | 87.8 | **90.2** |
| Walker2d | R | 0.1 | 0.1 | 0.3 | 17.5 | 13.4 | **21.7** | 13.3 | 12.4 | 13.1 | 4.1 | 3.0 | 4.3 | 21.8 | 21.8 | 21.9 |
| | M | 77.1 | 84.4 | **86.2** | 77.6 | 91.7 | **93.2** | 82.0 | 79.1 | **85.9** | 71.2 | 81.1 | **87.4** | 88.3 | 89.7 | **91.6** |
| | MR | 63.5 | 81.9 | **93.6** | 85.0 | 86.0 | 86.4 | 81.7 | 85.2 | **88.3** | 88.0 | 89.5 | **93.3** | 85.0 | 89.5 | **91.2** |
| | ME | 108.9 | 111.4 | 111.8 | 98.5 | 97.4 | **116.0** | 51.9 | 49.0 | **68.1** | 108.3 | 111.1 | **112.7** | 111.8 | 112.9 | **114.0** |
| | FR | 96.6 | 99.4 | **101.3** | 98.0 | 98.3 | **99.7** | 90.5 | 92.6 | **93.2** | 78.1 | 83.0 | **99.5** | 94.2 | 96.7 | **98.4** |

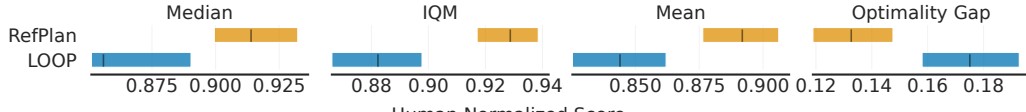

Figure 3: RLiable (Agarwal et al., 2022) comparison of RefPlan and LOOP. Across all four metrics (the higher the better for Median, IQM, and Mean, while the lower the better for Optimality Gap), RefPlan demonstrates superior performance with non-overlapping confidence intervals, highlighting statistically significant improvements over LOOP.

*HalfCheetah*, RefPlan outperformed the original policy. In *Walker2d*, RefPlan boosted performance by 16.4%, 31.4%, and 42.5% for COMBO, MAPLE, and CQL, respectively. Although the gains were more modest in *Hopper*, RefPlan still reduced performance drops. Overall, RefPlan showed strong resilience under high epistemic uncertainty caused by OOD initialization.

**4.2. RefPlan Enhances Any Offline-learned Policies**

To address RQ2, we evaluated the normalized score metric across the five offline policy learning algorithms. Table 2 shows that RefPlan outperformed baselines in 10 (CQL), 7 (EDAC), 12 (MOPO), 9 (COMBO), and 12 (MAPLE) of 15 tasks, matching performance in the others. Both MB planning methods, LOOP and RefPlan, improved performance, with RefPlan showing a more substantial gain. On average, RefPlan enhanced prior policy performance by 11.6%, compared to LOOP's 5.3%.

In order to make a more statistically rigorous comparison between RefPlan and LOOP, we leverage RLiable (Agarwal et al., 2022), a framework designed for robust evaluation of RL algorithms. RLiable focuses on statistically sound aggregate metrics, such as the median, interquartile mean (IQM), mean, and optimality gap, which provide a comprehensive view of algorithm performance across tasks. By

using bootstrapping with stratified sampling, RLiable also estimates confidence intervals, ensuring that comparisons are not skewed by outliers or noise.

We applied RLiable to compare RefPlan and LOOP across the tested environments and prior policy setups (Figure 3). Across all metrics, RefPlan consistently outperformed LOOP, with non-overlapping confidence intervals, indicating statistically significant improvements. These results demonstrate RefPlan's superior ability to enhance various offline policy learning algorithms by explicitly accounting for epistemic uncertainty during planning.

Another key question is whether the performance gains from RefPlan stem from its principled Bayesian framework or simply from a larger inference budget. To investigate this, we conducted an additional experiment using CQL as the prior policy on the MR and FR datasets. We compared RefPlan to LOOP under an equivalent computational load by allocating LOOP a 16-fold increase in its sampling budget, matching the maximum budget used by RefPlan (i.e., when $\bar{n} = 16$). The results, summarized in Table 3, demonstrate that while increasing the computational budget improves LOOP's performance, RefPlan consistently maintains its advantage across the tested configurations. This finding suggests that the superior performance of RefPlan is not merely a product of increased computation but is attributable

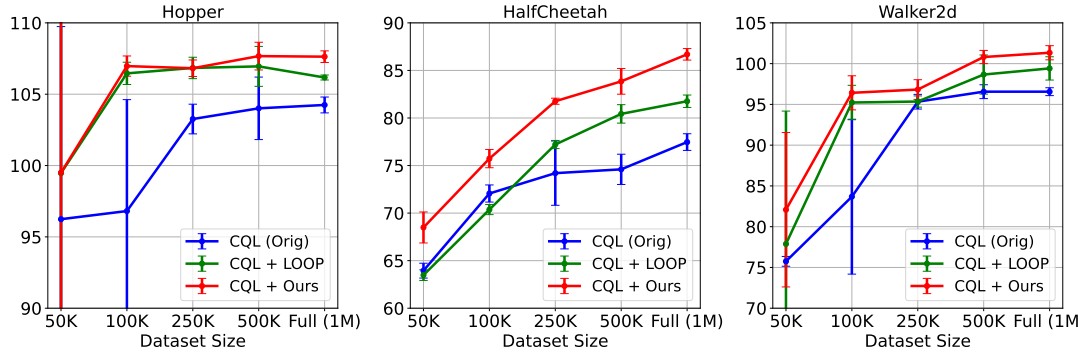

Figure 4: Performance comparison of RefPlan and LOOP across different dataset sizes in *Hopper*, *HalfCheetah*, and *Walker2d* environments using the FR dataset, which contains 1M samples. We use CQL as the prior policy learning algorithm, and the results represent the average and standard error calculated from three random seeds.

Table 3: RefPlan vs. LOOP with an equivalent inference budget.

| Environment | Config | LOOP (16x) | RefPlan |
|---|---|---|---|
| Hopper | MR | $97.8 \pm 1.1$ | $98.1 \pm 0.5$ |
| | FR | $107.5 \pm 0.6$ | $107.6 \pm 0.5$ |
| Walker2d | MR | $83.2 \pm 8.8$ | $93.6 \pm 1.1$ |
| | FR | $99.9 \pm 1.5$ | $101.3 \pm 0.3$ |
| HalfCheetah | MR | $53.2 \pm 0.1$ | $54.1 \pm 0.6$ |
| | FR | $83.1 \pm 0.8$ | $86.7 \pm 0.7$ |

Table 4: Average returns on *HalfCheetah* with dynamics changes.

| Task | Orig | LOOP | Ours |
|---|---|---|---|
| *joint* | 5295 | 6088 | **6190** |
| *hill* | 327.1 | 949.7 | **1224** |
| *gentle* | 1087 | 2363 | **2435** |
| *steep* | 2123 | 3245 | **6238** |
| *field* | 1205 | 2774 | **3345** |

to its explicit modeling and marginalization of epistemic uncertainty during planning.

### 4.3. Performance with Data with Different Sizes

With limited data, the agent faces increased epistemic uncertainty. A key question is whether RefPlan can better handle these scenarios with constrained data (RQ3). To explore this, we randomly subsample 50K, 100K, 250K, and 500K transition samples from the FR dataset for each environment. We then train the prior policy using CQL and compare its performance with that achieved when enhanced by either LOOP or RefPlan. As shown in Figure 4, RefPlan consistently demonstrates greater resilience to limited data, outperforming the baselines across all three environments.

### 4.4. Is RefPlan More Robust to Changing Dynamics?

To address RQ4, we evaluated RefPlan in the *HalfCheetah* environment under varying dynamics, including *disabled joint*, *hill*, slopes (*gentle* and *steep*), and *field*, following the approach of Clavera et al. (2019) (Appendix D). High epistemic uncertainty arises when dynamics differ from those seen during prior policy training. We trained the prior policy using the FR dataset, which contains the most diverse trajectories, and used MAPLE for its adaptive policy learning.

Table 4 shows that while MAPLE struggled with changed dynamics, MB planning methods improved performance. RefPlan achieved the best results across all variations but still faced notable drops, especially in the hill and gentle environments. Data augmentation for single-task offline RL could enhance adaptability, a topic for future work.

## 5. Related Work

**Offline RL** Policy distribution shift in offline RL leads to instabilities like extrapolation errors and value overestimation (Kumar et al., 2019; Fujimoto et al., 2019). To mitigate this, policy constraint methods limit deviation from the behavior policy (Wu et al., 2019; Kumar et al., 2019; Fujimoto & Gu, 2021), while value-based approaches penalize OOD actions (Kumar et al., 2020; An et al., 2021). Another strategy avoids querying OOD actions by learning values only from in-dataset samples and distilling a policy (Kostrikov et al., 2022).

MB offline policy learning trains a dynamics model from batch data to generate imaginary rollouts for dataset augmentation. To prevent exploiting model errors, ensemble-estimated uncertainty can be penalized in rewards (Yu et al., 2020; Kidambi et al., 2021; Lu et al., 2021). Alternatively, values of model-generated samples can be minimized (Yu et al., 2021), or adversarial dynamics models can discourage

selecting OOD actions (Rigter et al., 2022).

Offline policies are typically fixed after training, but Ghosh et al. (2021; 2022) show they can fail under high epistemic uncertainty, underscoring the need for adaptivity. APE-V (Ghosh et al., 2022) addresses this by using a value ensemble to approximate the distribution over environments, enabling policy adaptation during evaluation. MAPLE (Chen et al., 2021) employs an RNN to encode the agent's history into a dense vector for adaptive conditioning while leveraging an ensemble dynamics model to expose the policy to diverse simulated environments, improving robustness to uncertainty.

**Model-based planning for offline RL**  MB planning enhances responsiveness at test time. MBOP (Argenson & Dulac-Arnold, 2021) applies MPC with MPPI (Williams et al., 2015), adapting it for offline RL by using a BC policy for trajectory generation. Uncertain rollouts can be filtered based on ensemble disagreement (Zhan et al., 2022).

LOOP (Sikchi et al., 2021) improves offline-learned policies with planning, outperforming MBOP. It uses KL-regularized optimization for offline planning but only addresses epistemic uncertainty by penalizing ensemble variance in rewards during TrajOpt. In contrast, RefPlan takes a Bayesian approach, explicitly modeling epistemic uncertainty for better generalization and performance.

**Probabilistic interpretation of MB planning**  The control-as-inference framework (Levine, 2018; Abdolmaleki et al., 2018) provides a probabilistic view of control and RL problems, naturally leading to sampling-based solutions in MB planning (Piché et al., 2019). Okada & Taniguchi (2020) showed that various sampling-based TrajOpt algorithms can be derived from this perspective. Janner et al. (2022) introduced a diffusion-based planner using control-as-inference to derive a perturbation distribution, embedding reward signals into the diffusion sampling process. However, to our knowledge, we are the first to propose an offline MB planning algorithm that integrates an offline-learned policy as a prior in a Bayesian framework while explicitly modeling epistemic uncertainty within a unified probabilistic formulation.

**Bayesian RL and epistemic POMDP**  Bayesian RL (Ghavamzadeh et al., 2015) and BAMDPs (Duff, 2002) address learning optimal policies in unknown MDPs. A BAMDP can be reformulated as a belief-state MDP, where the belief serves as a sufficient statistic of the agent's history (Guez et al., 2012), framing BAMDPs as a special case of partially observable MDPs (POMDPs) (Kaelbling et al., 1998). Zintgraf et al. (2020) extended this perspective to meta-RL, introducing VariBAD, a variational inference-based method for approximating the belief distribution over possible environments.

Relatedly, Ghosh et al. (2021) introduced the *epistemic POMDP*, where an agent's epistemic uncertainty—arising from incomplete exploration or task ambiguity—induces partial observability. An epistemic POMDP is an instance of BAMDP, with a special focus on performance during a single evaluation episode rather than online learning and asymptotic regret minimization. Ghosh et al. (2022) further noted that offline RL in a single-task setting can be viewed as an epistemic POMDP, as static offline datasets typically cover only a subset of the state-action space, leading to partial observability in environment dynamics beyond the dataset.

We also adopt the epistemic POMDP perspective for single-task offline RL but focus on MB planning. Unlike prior approaches, our goal is to enhance offline-learned policies by addressing epistemic uncertainty, enabling more effective generalization during deployment.

## 6. Conclusion

In this paper, we introduced **RefPlan** (Reflect-then-Plan), a novel *doubly Bayesian* approach to offline model-based planning that integrates epistemic uncertainty modeling with model-based planning in a unified probabilistic framework. Our method enhances offline RL by explicitly accounting for epistemic uncertainty, a common challenge in offline settings where data coverage is often incomplete. Through extensive experiments on standard offline RL benchmarks, we demonstrated that RefPlan consistently outperforms existing methods, particularly under challenging conditions of OOD initialization, limited data availability, and changing environment dynamics, making it a valuable tool for more reliable and adaptive offline RL. Future work could extend RefPlan to more complex models and environments.

## Acknowledgements

This work was supported by the Institute of Information & Communications Technology Planning & Evaluation (IITP) grant funded by the Korean Government (MSIT) (No. RS-2024-00457882, National AI Research Lab Project).

## Impact Statement

This paper presents work whose goal is to advance the field of Machine Learning, particularly Reinforcement Learning. There are many potential societal consequences of our work, none which we feel must be specifically highlighted here.

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

# A. Additional Background

## A.1. Bayes-Adaptive Markov Decision Processes

Bayes-Adaptive Markov Decision Processes (BAMDPs) (Duff, 2002) extend the standard MDP framework by explicitly incorporating uncertainty over the transition and reward functions. In a BAMDP, instead of assuming that the transition dynamics $T(\mathbf{s}'|\mathbf{s}, \mathbf{a})$ and reward function $r(\mathbf{s}, \mathbf{a})$ are known and fixed, we assume that they are drawn from an unknown distribution. The agent maintains a posterior belief over these functions and updates it as new data are collected through interaction with the environment.

To illustrate, consider a simple case where we have finite and discrete state and action spaces with $|\mathcal{S}| = n_s$ and $|\mathcal{A}| = n_a$; hence, a state can be represented with an integer, i.e., $s = i$ for $i = 1, \ldots, n_s$, and similarly for the actions. While the reward function $r(s, a)$ is assumed to be known, we are uncertain about the transition probabilities $T(s'|s, a)$. We can model this uncertainty by placing a prior distribution over the transition probabilities, typically using a Dirichlet prior, which is conjugate to the multinomial likelihood of observing transitions between states.

For each state-action pair $(s, a) \in \mathcal{S} \times \mathcal{A}$, the transition probabilities $T(s'|s, a)$ are parameterized by a multinomial distribution:

$$T(s'|s, a) \sim \text{Multinomial}(\boldsymbol{\theta}_{s,a,s'}), \tag{11}$$

where $\boldsymbol{\theta}_{s,a} = (\boldsymbol{\theta}_{s,a,1}, \ldots, \boldsymbol{\theta}_{s,a,n_s})$ represents the probabilities of transitioning from state $s$ to any state $s' \in \mathcal{S}$ under action $a$. These parameters follow a Dirichlet distribution:

$$\boldsymbol{\theta}_{s,a} \sim \text{Dirichlet}(\alpha_{s,a}), \tag{12}$$

where $\alpha_{s,a} = (\alpha_{s,a,1}, \ldots, \alpha_{s,a,n_s}) > 0$ are the Dirichlet hyperparameters.

Initially, the agent holds a prior belief about the transition probabilities, represented by the Dirichlet hyperparameters $\alpha_{s,a}$ for all state-action pairs. As the agent interacts with the environment and observes transitions of the form $(s, a, s')$, it updates its posterior belief by simply updating the corresponding Dirichlet hyperparameters. Specifically, when the agent observes a transition from state $s$ to state $s'$ under action $a$, the corresponding Dirichlet hyperparameter is updated as:

$$\alpha_{s,a,s'} \leftarrow \alpha_{s,a,s'} + 1, \tag{13}$$

while all other Dirichlet hyperparameters remain unchanged. This process of updating the Dirichlet hyperparameters fully captures the agent's experiences; hence, these hyperparameters act as sufficient statistics for the agent's belief about the environment.

By transforming the BAMDP into a belief-state MDP, where the belief state $b_t = p(\boldsymbol{\theta}|\tau_{:t})$ is a distribution over transition probabilities conditioned on the observed trajectory $\tau_{:t} = (s_0, a_0, s_1, \ldots, s_t)$, the agent can solve the problem using standard MDP solution methods. The augmented state space, or hyper-state space, includes both the *physical* state $s \in \mathcal{S}$ and the belief state $b \in \mathcal{B}$. In this simple finite state-action example, the belief state corresponds to the Dirichlet hyperparameters $\alpha$.

The transition dynamics of the resulting belief-state MDP are fully known and can be written as:

$$T(\bar{s}'|\bar{s}, a) = T(s', \alpha'|s, \alpha, a) = T(s'|s, a, \alpha)p(\alpha'|s, \alpha, a) \tag{14}$$

$$= \frac{\alpha_{s,a,s'}}{\sum_{s'' \in \mathcal{S}} \alpha_{s,a,s''}} \mathbb{I}\left(\alpha'_{s,a,s'} = \alpha_{s,a,s'} + 1\right), \tag{15}$$

where $\mathbb{I}(\cdot)$ is the indicator function. This transformation turns the BAMDP into a fully observable MDP in the hyper-state space, which allows the use of standard, e.g., DP methods to compute an optimal policy.

However, the computational complexity of solving the BAMDP grows quickly with the number of states and actions. If the states are fully connected (i.e., $p(s'|s, a) > 0, \ \forall s, a, s'$), the number of reachable belief states increases exponentially over time, making exact solutions intractable for even moderately sized problems.

For a comprehensive overview of solution methods for BAMDPs, we refer readers to the survey by Ghavamzadeh et al. (2015). In this work, we have utilized variational inference techniques from Zintgraf et al. (2020) and Dorfman et al. (2021) to approximate the agent's posterior belief over the environment dynamics, $p(b|\tau_{:t})$, based on past experiences.

Table 5: Performance comparison of RefPlan against baseline methods on Hopper, HalfCheetah, and Walker2d tasks using MOPO and COMBO for offline policy optimization. The table evaluates original policies (Orig), policies trained with Non-Markovian (NM) dynamics models (NM (Train)), NM-trained policies combined with RefPlan for planning (NM (Train) + RefPlan), and RefPlan using original policies as priors. Results demonstrate RefPlan's ability to improve test-time performance across different dynamics models and environments.

| | | Orig | NM (Train) | NM (Train) +RefPlan | RefPlan |
|---|---|---|---|---|---|
| **Hopper** | M | 66.9 | - | - | **67.7** |
| | MR | 90.3 | 93.2 | **98.18** | 94.5 |
| | ME | 91.3 | - | - | **96.5** |
| **HalfCheetah** | M | 42.8 | 40.6 | **66.45** | 59.8 |
| | MR | 70.6 | 53.2 | 72.46 | **73.8** |
| | ME | 73.5 | 71.6 | **100.34** | 96.6 |
| **Walker2d** | M | 82.0 | 60.6 | 72.73 | **85.9** |
| | MR | 81.7 | 53.3 | 79.75 | **88.3** |
| | ME | 51.9 | 42.4 | 64.59 | **68.1** |

| | | Orig | NM (Train) | NM (Train) +RefPlan | RefPlan |
|---|---|---|---|---|---|
| **Hopper** | M | 60.9 | 52.2 | 62.30 | **77.2** |
| | MR | 101.1 | 44.9 | 61.90 | **101.8** |
| | ME | 105.6 | 27.3 | 39.23 | **107.8** |
| **HalfCheetah** | M | 67.2 | 30.3 | 41.61 | **77.4** |
| | MR | 73.0 | 47.6 | 59.54 | **75.0** |
| | ME | 97.6 | 93.5 | 109.25 | **110.3** |
| **Walker2d** | M | 71.2 | 79.1 | **89.43** | 87.4 |
| | MR | 88.0 | 80.4 | 91.01 | **93.3** |
| | ME | 108.3 | 36.7 | 38.47 | **112.7** |

# B. Additional Results

## B.1. Performance comparison: non-Markovian dynamics model for training vs. planning

The experiments presented in Table 5 aims to evaluate the effectiveness of RefPlan in leveraging the VAE dynamics—consisting of the variational encoder $q_\phi$ and the probabilistic ensemble decoder $\hat{p}_\psi$ (Figure 6)—for planning at test time. Specifically, these experiments compare the following approaches:

- "Orig": the original prior policy trained using MOPO or COMBO.

- "NM (Train)": the policy trained using a non-Markovian (NM) VAE dynamics model during offline policy optimization via MOPO or COMBO.

- "NM (Train) + RefPlan ": the RefPlan agent that uses the policies trained using NM dynamics models as priors.

- "RefPlan ": the RefPlan agent that uses the original prior policies as priors.

The results demonstrate several key findings. First, RefPlan consistently outperforms NM (Train) across all environments and datasets, confirming that the VAE dynamics models are significantly more effective when used for planning at test time rather than during offline policy training. This highlights RefPlan's ability to explicitly handle epistemic uncertainty, leveraging the agent's real-time history to infer the underlying MDP dynamics.

Second, in MOPO results, NM (Train) diverged or underperformed in several cases. This suggests that the heuristically estimated model uncertainty used in MOPO is not well-suited for integrating with the VAE dynamics models during offline training. Even with large penalty parameters, the value function diverged in the Hopper tasks, indicating a fundamental limitation in using NM models with MOPO for policy optimization. By contrast, COMBO results did not exhibit these issues, suggesting that COMBO's framework is better equipped to incorporate such dynamics models during training.

Finally, applying RefPlan to policies trained with NM dynamics models (NM (Train) + RefPlan) further boosted test-time performance, often by substantial margins. This demonstrates that even when NM dynamics models introduce suboptimality during offline training, RefPlan can recover and enhance the policy's performance through effective planning at test time. Across all environment-dataset combinations, RefPlan provides robust improvements over both the original and NM (Train)-optimized policies, further validating its capability to address epistemic uncertainty and improve the generalization of offline-learned policies.

## B.2. Evaluating the impact of the number of latent samples on variances and performance

This experiment evaluates how the sample variance of the marginal action posterior mean from (10) changes with the number of latent samples ($\bar{n}$) used in the outer expectation. At each time step, we compute the posterior mean $K$ times, calculate its

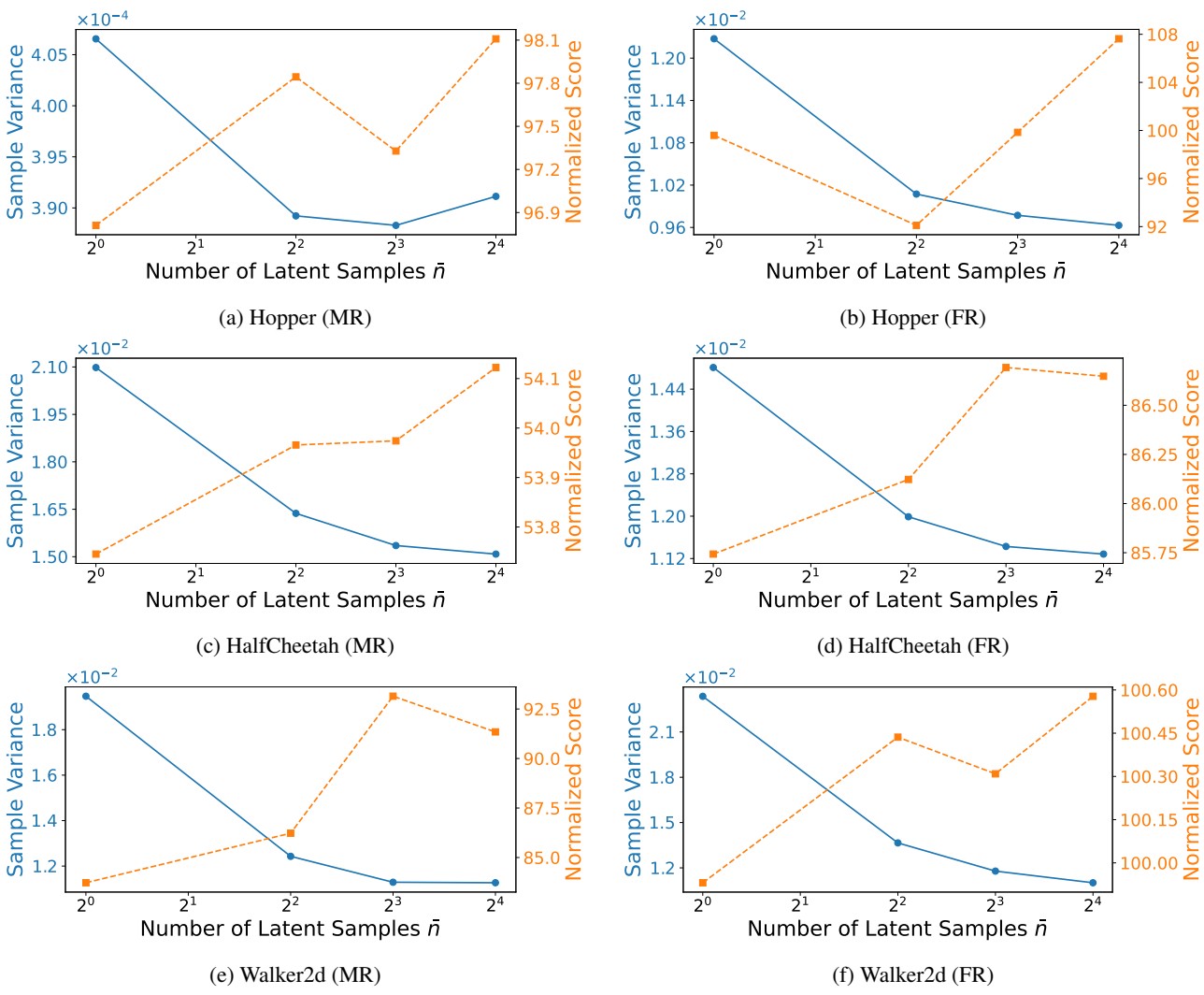

Figure 5: The sample variance and the performance vs. the number of latent samples of RefPlan, evaluated from three environments with the MR and FR datasets using CQL as the prior policy.

variance averaged across action dimensions, and report the running average over a 1,000-step episode. Results are averaged over three random seeds, with CQL as the prior policy, across three environments (Hopper, HalfCheetah, Walker2d) and two dataset configurations (MR, FR).

The figures show that as $\bar{n}$ increases, the average sample variance decreases, with $\bar{n} = 1$ consistently yielding the highest variance. Performance, measured as normalized scores, generally improves with increasing $\bar{n}$, suggesting a positive correlation between reduced variance and higher performance. However, while reduced variance likely contributes to this improvement, further investigation is needed to confirm causality, as other factors may also play a role.

## C. Algorithm Details

### C.1. Algorithm Summary

RefPlan is designed to enhance any offline RL policy by incorporating MB planning that accounts for epistemic uncertainty. The algorithm operates in two primary stages: pretraining (Appendix C.3) and test-time planning.

---

**Algorithm 1** Offline MB Planning

---

**Input:** $\hat{p}$, $V_\phi$, $\mathcal{D}$, $\pi_\theta$, $\mathfrak{L}$
Train $\hat{p}_\psi$ with $\mathcal{D}$ via MLE
Train $V_\phi$ and $\pi_\theta$ with $\mathfrak{L}$ and $\mathcal{D}$
$t \leftarrow 1$
**repeat**
 Observe $\mathbf{s}_t$
 $\mathbf{a}^*_{t:t+H} \leftarrow \text{TrajOpt}(\mathbf{s}_t, \hat{p}_\psi, \pi_\theta, V_\phi)$
 Take $\mathbf{a}^*_t$, observe $\mathbf{s}_{t+1}, r_t$
 $t \leftarrow t + 1$
**until** episode terminates

---

**Algorithm 2** RefPlan: Offline MB Planning as Probabilistic Inference

---

**Input:** $\tau_{:t} = (\mathbf{s}_0, \mathbf{a}_0, r_0, \ldots, \mathbf{s}_t)$, $\hat{p}_\psi$, $q_\phi$, $\pi_\mathrm{p}$, $\hat{Q}$, $H$, $\bar{N}$, $\bar{n}$, $\kappa$
$\mu_t$, $\sigma_t \leftarrow q_\phi(\cdot|\tau_{:t})$ {Get the Gaussian parameters}
$\{m_t^j\}_{j=1}^{\bar{n}} \sim \mathcal{N}(\mu_t, \sigma_t^2)$ {Sample $\bar{n}$ latent vectors from the approximate posterior}
**for** $n = 1, \ldots, \bar{N}$ **do**
 **for** $h = 0, \ldots, H-1$ **do**
  $\mathbf{a}_{t+h}^n \sim \pi_\mathrm{p}(\cdot|\mathbf{s}_{t+h})$ {Sample prior action sequence}
  $\mathbf{s}_{t+h+1}^n \sim \hat{p}_\psi(\cdot|\mathbf{s}_{t+h}^n, \mathbf{a}_{t+h}^n, \mu_t)$ {Sample the next state from model using $\mu_t$}
 **end for**
 **for** $j = 1, \ldots, \bar{n}$ **do**
  $\mathbf{s}_t^{n,j} \leftarrow \mathbf{s}_t$
  **for** $h = 0, \ldots, H-1$ **do**
   $\mathbf{s}_{t+h+1}^{n,j} \sim \hat{p}_\psi(\cdot|\mathbf{s}_{t+h}^{n,j}, \mathbf{a}_{t+h}^n, m_t^j)$ {Sample next state from model using $m_t^j$}
   $r_{t+h}^{n,j} \leftarrow r(\mathbf{s}_{t+h}^{n,j}, \mathbf{a}_{t+h}^n, m_t^j)$ {Compute the reward using $m_t^j$}
  **end for**
 **end for**
**end for**
Compute $\mathbb{E}_{p(\tau|\mathcal{O})}[\mathbf{a}_{t:t+H}]$ with (10)
**return** $\mathbb{E}_{p(\tau|\mathcal{O})}[\mathbf{a}_{t:t+H}]$ {Return the plan to be used in line 7 of Algorithm 1}

---

**Pretraining stage**  The first step is to train a prior policy $\pi_\mathrm{p}$ using any offline RL algorithm. In parallel, a VAE is trained using the ELBO objective in (8), where the encoder captures the agent's epistemic uncertainty and the decoder learns the environment dynamics. See Appendix C.3 for more details.

**Test-time planning stage**  During evaluation, the agent employs MPC (Algorithm 1), where RefPlan serves as the trajectory optimization subroutine. At each time step $t$, the agent gathers its history $\tau_{:t}$ and encodes it into a latent variable $m_t$ using the pretrained encoder (line 2 of Algorithm 2). This latent variable encapsulates the agent's current belief about the environment, reflecting epistemic uncertainty.

Then, we first generate $\bar{N}$ prior plans with the prior policy and the learned model (lines 5-8). Each plan has the length of $H$, and we use $\mu_t$ to condition $\hat{p}_\psi$ at this stage. Optionally, we add a Gaussian noise to the actions sampled by $\pi_\mathrm{p}$, following Argenson & Dulac-Arnold (2021); Sikchi et al. (2021).

Once the prior plans are prepared, we rollout the plans under the learned model to generate multiple trajectories. That is, for each sampled $m_t$, we obtain $\bar{N}$ trajectories (lines 9-14). These trajectories are then used to estimate the optimal plan, conditioned on $m_t^j$. We marginalize out the latent variable via Monte-Carlo expectation using the law of total expectation.

Finally, the first action from the optimized plan is selected and executed in the environment. This process repeats at each subsequent time step, with the agent continuously updating its belief state and re-optimizing its plan based on new observations.

Table 6: Hyperparameters for Model Architecture and Training

| Architecture Hyperparameters | Value |
|---|---|
| Task Embedding Dimension | 16 |
| State Embedding Dimension | 16 |
| Action Embedding Dimension | 16 |
| Reward Embedding Dimension | 4 |
| GRU Hidden Dimension | 256 |
| Decoder Network Architecture | Fully connected, [200, 200, 200, 200] with skip connection |
| Decoder ensemble size | 20 |
| Decoder number of elite models | 14 |
| **Training Hyperparameters** | **Value** |
| KL Weight Coefficient | 0.1 |
| Input Normalization | True |
| Learning Rate | 0.001 |
| Weight Decay | 0.01 |
| Optimizer | AdamW |
| Batch Size | 64 |

### C.2. Architecture

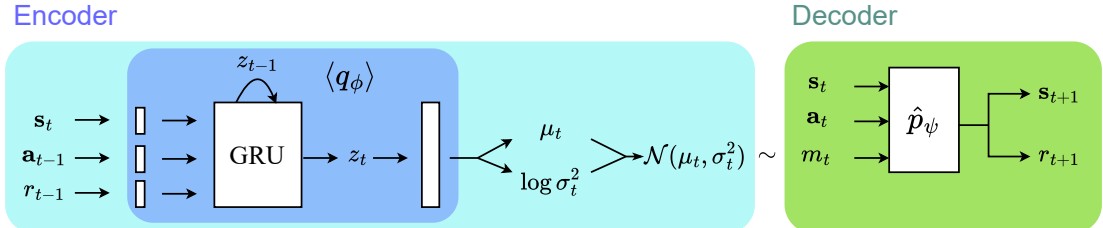

Figure 6: A schematic illustration of the architecture of RefPlan. We use the same encoder architecture as in VariBAD (Zintgraf et al., 2020), which consists of a GRU model and a fully connected layer. Unlike VariBAD, which uses the decoder only for training the encoder, we employ a two-stage training procedure (Appendix C.3) to learn a decoder that is directly used for planning at test time. The decoder network reconstructs the past trajectory and predicts the next state but does not attempt to predict the entire future trajectory as in the prior work (see also Eq.(9)).

Figure 6 illustrates the architecture of RefPlan. For the encoder, we adopt the architecture from VariBAD (Zintgraf et al., 2020), with a few minor modifications to the hyperparameters. The encoder utilizes a GRU network to encode the agent's history and outputs the parameters of a Gaussian distribution representing the latent variable $m_t$.

At time $t = 0$, we initialize $z_{-1} = 0$ and $\mathbf{a}_{-1} = 0$. The state $\mathbf{s}_t$, the previous action $\mathbf{a}_{t-1}$, and the previous reward $r_{t-1}$ are first embedded into their respective latent spaces using distinct linear layers, each followed by ReLU activation. These embedded vectors, along with the hidden state from the previous time step $z_{t-1}$, are then processed by the GRU, which outputs the updated hidden state $z_t$. This hidden state is subsequently linearly projected onto the task embedding space to obtain the mean ($\mu_t$) and log variance ($\log \sigma_t^2$) of the Gaussian distribution for the latent variable at the current time step.

Since the decoder plays a critical role in test-time planning, we follow established practices from prior work and implement the decoder using a probabilistic ensemble network (Chua et al., 2018; Janner et al., 2019; Yu et al., 2020; 2021; Chen et al., 2021). Specifically, the ensemble consists of 20 models, from which we select the 14 *elite* models that achieve the lowest validation loss during training. The decoder network conditions on a latent sample $m_t \sim \mathcal{N}(\mu_t, \sigma_t^2)$, along with $\mathbf{s}_t$ and $\mathbf{a}_t$, to predict the next state $\mathbf{s}_{t+1}$ and reward $r_{t+1}$. The hyperparameters associated with the architecture are summarized in Table 6.

Table 7: Reproducing the reported performances of offline policy learning algorithms on the D4RL MuJoCo tasks. *Numbers reported in An et al. (2021).

| | | CQL | | EDAC | | MOPO | | COMBO | | MAPLE | |
|---|---|---|---|---|---|---|---|---|---|---|---|
| | | Paper | Rep. | Paper | Rep. | Paper | Rep. | Paper | Rep. | Paper | Rep. |
| Hopper | R | 10.8 | 1.0 | 25.3 | 23.6 | 11.7 | 32.2 | 17.9 | 6.3 | 10.6 | 31.5 |
| | M | 86.6 | 66.9 | 101.6 | 101.5 | 28.0 | 66.9 | 97.2 | 60.9 | 21.1 | 29.4 |
| | MR | 48.6 | 94.6 | 101.0 | 100.4 | 67.5 | 90.3 | 89.5 | 101.1 | 87.5 | 61.0 |
| | ME | 111.0 | 111.4 | 110.7 | 106.7 | 23.7 | 91.3 | 111.1 | 105.6 | 42.5 | 46.9 |
| | FR | 101.9* | 104.2 | 105.4 | 106.6 | - | 73.2 | - | 89.9 | - | 79.1 |
| HalfCheetah | R | 35.4 | 19.9 | 28.4 | 22.5 | 35.4 | 29.8 | 38.8 | 40.3 | 38.4 | 33.5 |
| | M | 44.4 | 47.4 | 65.9 | 63.8 | 42.3 | 42.8 | 54.2 | 67.2 | 50.4 | 68.8 |
| | MR | 46.2 | 47.0 | 61.3 | 61.8 | 53.1 | 70.6 | 55.1 | 73.0 | 59.0 | 71.5 |
| | ME | 62.4 | 98.3 | 106.3 | 100.8 | 63.3 | 73.5 | 90.0 | 97.6 | 63.5 | 64.0 |
| | FR | 76.9* | 77.5 | 84.6 | 81.7 | - | 81.7 | - | 71.8 | - | 66.8 |
| Walker2d | R | 7.0 | 0.1 | 16.6 | 17.5 | 13.6 | 13.3 | 7.0 | 4.1 | 21.7 | 21.8 |
| | M | 74.5 | 77.1 | 92.5 | 77.6 | 17.8 | 82.0 | 81.9 | 71.2 | 56.3 | 88.3 |
| | MR | 32.6 | 63.5 | 87.1 | 85.0 | 39.0 | 81.7 | 56.0 | 88.0 | 76.7 | 85.0 |
| | ME | 98.7 | 108.9 | 114.7 | 98.5 | 44.6 | 51.9 | 103.3 | 108.3 | 73.8 | 111.8 |
| | FR | 94.2* | 96.6 | 99.8 | 98.0 | - | 90.5 | - | 78.1 | - | 94.2 |

### C.3. Pretraining

RefPlan requires two stages of pretraining. First, we use an off-the-shelf offline RL algorithm to train a prior policy $\pi_{\mathrm{p}}$. In our experiments, we evaluated several algorithms, including CQL (Kumar et al., 2020), EDAC (An et al., 2021), MOPO (Yu et al., 2020), COMBO (Yu et al., 2021), and MAPLE (Chen et al., 2021), though any offline RL policy learning algorithm could be utilized.

Second, we train the encoder $q_\phi$ and the decoder $\hat{p}_\psi$. The encoder $q_\phi$ is trained using the ELBO loss as defined in (8). The decoder $\hat{p}_\psi$ is trained to reconstruct the past and to predict the next state, conditioned on the sample $m_t$ the current state $\mathbf{s}_t$, and the action $\mathbf{a}_t$. This training constitutes the first phase of dynamics learning. During this step, the encoder learns a latent representation that captures essential information for reconstructing the trajectory. Unlike VariBAD, where the decoder is trained to reconstruct the entire trajectory including future states, we found that focusing on the past and the next state improves the decoder's performance.

After completing the first training phase, we freeze the encoder network parameters and proceed to the second phase. In this phase, we fine-tune the decoder network $\hat{p}_\psi$ to accurately predict the next state given $m_t$, $\mathbf{s}_t$, and $\mathbf{a}_t$. This is achieved using the loss function that we reiterate here for clarity:

$$L(\psi) = \mathbb{E}_{\tau \sim \mathcal{D}} \left[ \sum_{h=0}^{H-1} \mathbb{E}_{m_h \sim q_\phi(\cdot | \tau_{:h})} [-\log \hat{p}_\psi(\mathbf{s}_{h+1}, r_h | \mathbf{s}_h, \mathbf{a}_h, m_h)] \right]. \tag{16}$$

The second training phase ensures that the learned dynamics model, $\hat{p}_\psi$, accurately predicts the next state. This two-stage approach allows RefPlan to maintain an effective dynamics moel for planning at test time, unlike VariBAD, where the decoder is discarded after training the VAE.

## D. Experimental Details

### D.1. Experimental settings

**D4RL MuJoCo environments & datasets**   We use the v2 version for each dataset as provided by the D4RL library (Fu et al., 2020).

**Evaluation under high epistemic uncertainty due to OOD initialization (RQ1)**   To address RQ1, we assessed a policy trained on the ME dataset of each MuJoCo environment by initializing the agent from a state randomly selected from the R dataset. The results, presented in Figure 1, 1, and 1 are averaged over 3 seeds. For a fair comparison, the same initial state was used across all methods being compared—the prior policy, LOOP, and RefPlan—under the same random seed.

**Benchmarking on D4RL tasks (RQ2)**   To generate the benchmark results shown in Table 2, we first trained the five baseline policies on each dataset across the three environments. The focus of our analysis is on the performance

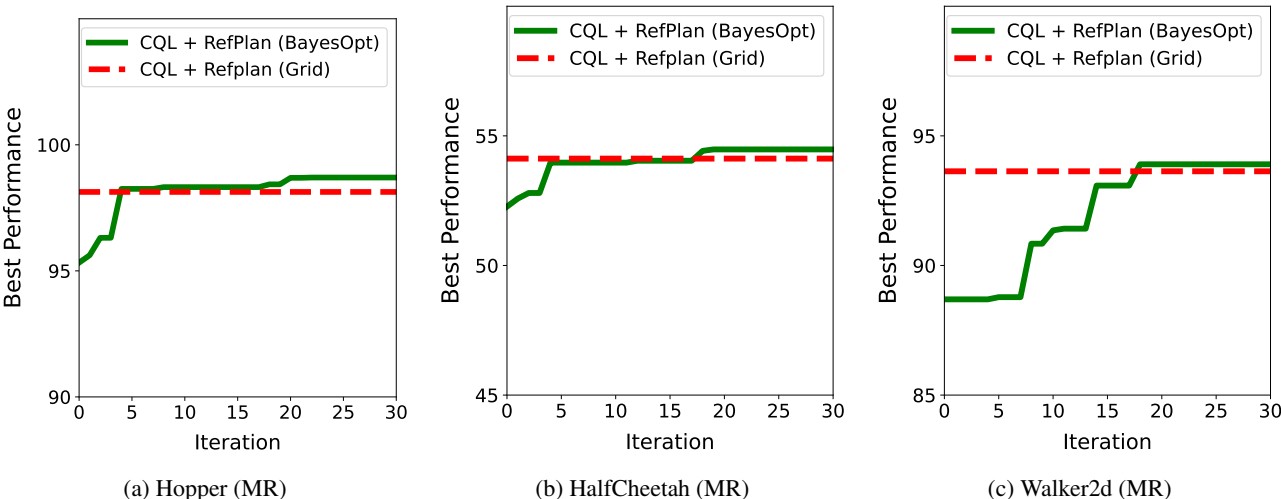

Figure 7: Best performance vs. the number of BayesOpt iterations, using CQL as a prior policy on the MR datasets across three environments.

improvements of these prior policies when augmented with either LOOP or RefPlan as an MB planning algorithm during the evaluation phase. *Thus, our approach is designed to be complementary to any offline policy learning algorithms, making the relative performance gains more relevant than the absolute performance of each algorithm.* Nevertheless, we aimed to closely replicate the original policy performance reported in prior studies. Table 7 compares our reproduced results with those originally reported. Overall, our implementation closely matches the original performances, often exceeding them significantly across various datasets. However, in some cases, our reproduced policy checkpoints underperformed compared to the originally reported results, such as CQL on the R datasets, EDAC on Walker2d M and ME datasets, COMBO on the Hopper R and M datasets, and MAPLE on the Hopper MR dataset. We will make our code publicly available upon acceptance.

**Varying dataset sizes (RQ3)**    In Figure 4, we present the normalized average return scores for CQL and its enhancements with either LOOP or RefPlan as we vary the dataset size from 50K to 500K. We conducted these experiments using the FR dataset across three environments, which originally contains 1M transition samples. To create the smaller datasets, we randomly subsampled trajectories. If the subsampled data exceeded the desired dataset size, we trimmed the last trajectory accordingly. For CQL training, we applied the same hyperparameters as those used for the full FR dataset.

**Changing dynamics (RQ4)**    To explore RQ4, we adapted the HalfCheetah environment following the approach of Clavera et al. (2019), introducing five variations: *disabled joint*, *hill*, *gentle* slope, *steep* slope, and *field*. These variations were implemented using the code from `https://github.com/iclavera/learning_to_adapt`. Unlike the original work, which focuses on meta-RL, our study addresses an offline RL problem within a single task framework. Hence, to make the tasks easier, we modified the `height` parameter for most variations, excluding the *disabled joint* task. The specific adjustments to the height parameters are detailed in Table 8. These changes were intended to create more manageable tasks while still providing a meaningful challenge for the offline RL algorithms.

Table 8: Environment configuration for the HalfCheetah variations used in RQ4 experiments, showing the original and modified `height` parameter values for each task.

| Task | Original | Modified |
|------|----------|----------|
| *hill* | 0.6 | 0.2 |
| *gentle* | 1 | 0.2 |
| *steep* | 4 | 0.5 |
| *field* | $Uniform(0.2, 1)$ | $Uniform(0.05, 0.4)$ |

### D.2. Hyperparameters

Table 9-13 outline the hyperparameters used for RefPlan across the five prior policies discussed in Section 4. We conducted a grid search over the following hyperparameters: the planning horizon $H \in \{2, 4\}$, the standard deviation of the Gaussian

Table 9: Hyperparameters used for MAPLE + RefPlan used on D4RL MuJoCo Gym environments

| | Hopper | | | | | HalfCheetah | | | | | Walker2D | | | | |
|---|---|---|---|---|---|---|---|---|---|---|---|---|---|---|---|
| | $H$ | $\sigma$ | $\kappa$ | $\bar{n}$ | $p$ | $H$ | $\sigma$ | $\kappa$ | $\bar{n}$ | $p$ | $H$ | $\sigma$ | $\kappa$ | $\bar{n}$ | $p$ |
| random | 2 | 0.01 | 10.0 | 1 | 1.0 | 4 | 0.05 | 10.0 | 16 | 1.0 | 2 | 0.05 | 0.1 | 16 | 1.0 |
| medium | 2 | 0.01 | 0.1 | 8 | 0.1 | 4 | 0.01 | 5.0 | 16 | 0.5 | 2 | 0.05 | 10.0 | 1 | 0.1 |
| med-replay | 4 | 0.01 | 5.0 | 1 | 0.1 | 4 | 0.01 | 10.0 | 8 | 0.1 | 4 | 0.05 | 0.1 | 16 | 0.1 |
| med-expert | 2 | 0.01 | 0.1 | 1 | 1.0 | 4 | 0.01 | 10.0 | 4 | 0.1 | 2 | 0.01 | 10.0 | 8 | 0.5 |
| full-replay | 2 | 0.01 | 10.0 | 1 | 0.5 | 2 | 0.01 | 5.0 | 16 | 0.1 | 2 | 0.05 | 5.0 | 1 | 1.0 |

Table 10: Hyperparameters used for COMBO + RefPlan used on D4RL MuJoCo Gym environments

| | Hopper | | | | | HalfCheetah | | | | | Walker2d | | | | |
|---|---|---|---|---|---|---|---|---|---|---|---|---|---|---|---|
| | $H$ | $\sigma$ | $\kappa$ | $\bar{n}$ | $p$ | $H$ | $\sigma$ | $\kappa$ | $\bar{n}$ | $p$ | $H$ | $\sigma$ | $\kappa$ | $\bar{n}$ | $p$ |
| random | 2 | 0.05 | 0.1 | 16 | 0.1 | 2 | 0.05 | 0.1 | 4 | 0.1 | 4 | 0.01 | 0.5 | 16 | 1.0 |
| medium | 2 | 0.01 | 10.0 | 16 | 0.5 | 4 | 0.05 | 5.0 | 16 | 0.1 | 4 | 0.05 | 1.0 | 16 | 0.5 |
| med-replay | 4 | 0.01 | 0.5 | 8 | 1.0 | 2 | 0.01 | 5.0 | 4 | 0.1 | 4 | 0.01 | 5.0 | 16 | 0.1 |
| med-expert | 4 | 0.01 | 0.5 | 8 | 1.0 | 2 | 0.05 | 5.0 | 16 | 0.1 | 4 | 0.01 | 10.0 | 16 | 0.1 |
| full-replay | 4 | 0.01 | 0.1 | 8 | 0.5 | 4 | 0.01 | 10.0 | 4 | 0.1 | 4 | 0.05 | 1.0 | 8 | 0.5 |

Table 11: Hyperparameters used for MOPO + RefPlan used on D4RL MuJoCo Gym environments

| | Hopper | | | | | HalfCheetah | | | | | Walker2d | | | | |
|---|---|---|---|---|---|---|---|---|---|---|---|---|---|---|---|
| | $H$ | $\sigma$ | $\kappa$ | $\bar{n}$ | $p$ | $H$ | $\sigma$ | $\kappa$ | $\bar{n}$ | $p$ | $H$ | $\sigma$ | $\kappa$ | $\bar{n}$ | $p$ |
| random | 2 | 0.01 | 5.0 | 1 | 0.1 | 4 | 0.01 | 10.0 | 8 | 0.1 | 2 | 0.05 | 0.1 | 8 | 1.0 |
| medium | 2 | 0.05 | 5.0 | 1 | 0.1 | 4 | 0.05 | 10.0 | 4 | 0.1 | 2 | 0.05 | 5.0 | 4 | 1.0 |
| med-replay | 4 | 0.05 | 5.0 | 16 | 0.1 | 2 | 0.05 | 10.0 | 4 | 0.1 | 4 | 0.01 | 1.0 | 16 | 0.1 |
| med-expert | 4 | 0.05 | 1.0 | 8 | 0.1 | 4 | 0.01 | 10.0 | 16 | 0.1 | 4 | 0.01 | 10.0 | 16 | 1.0 |
| full-replay | 2 | 0.05 | 5.0 | 16 | 1.0 | 4 | 0.05 | 5.0 | 16 | 0.1 | 4 | 0.05 | 10.0 | 1 | 0.1 |

Table 12: Hyperparameters used for CQL + RefPlan used on D4RL MuJoCo Gym environments

| | Hopper | | | | | HalfCheetah | | | | | Walker2d | | | | |
|---|---|---|---|---|---|---|---|---|---|---|---|---|---|---|---|
| | $H$ | $\sigma$ | $\kappa$ | $\bar{n}$ | $p$ | $H$ | $\sigma$ | $\kappa$ | $\bar{n}$ | $p$ | $H$ | $\sigma$ | $\kappa$ | $\bar{n}$ | $p$ |
| random | 4 | 0.01 | 10.0 | 1 | 0.1 | 4 | 0.05 | 5.0 | 1 | 0.1 | 4 | 0.05 | 10.0 | 16 | 1.0 |
| medium | 2 | 0.01 | 5.0 | 16 | 0.5 | 2 | 0.01 | 5.0 | 1 | 0.5 | 2 | 0.05 | 10.0 | 16 | 1.0 |
| med-replay | 4 | 0.05 | 0.1 | 8 | 0.1 | 4 | 0.01 | 5.0 | 16 | 0.1 | 4 | 0.05 | 1.0 | 4 | 0.1 |
| med-expert | 4 | 0.01 | 1.0 | 1 | 0.5 | 2 | 0.01 | 5.0 | 8 | 0.1 | 2 | 0.01 | 5.0 | 8 | 0.1 |
| full-replay | 4 | 0.05 | 10.0 | 16 | 0.1 | 2 | 0.01 | 5.0 | 8 | 0.1 | 4 | 0.01 | 5.0 | 8 | 0.1 |

Table 13: Hyperparameters used for EDAC + RefPlan used on D4RL MuJoCo Gym environments

| | Hopper | | | | | HalfCheetah | | | | | Walker2d | | | | |
|---|---|---|---|---|---|---|---|---|---|---|---|---|---|---|---|
| | $H$ | $\sigma$ | $\kappa$ | $\bar{n}$ | $p$ | $H$ | $\sigma$ | $\kappa$ | $\bar{n}$ | $p$ | $H$ | $\sigma$ | $\kappa$ | $\bar{n}$ | $p$ |
| random | 4 | 0.05 | 10.0 | 16 | 0.5 | 4 | 0.05 | 5.0 | 8 | 0.1 | 2 | 0.05 | 10.0 | 4 | 0.1 |
| medium | 2 | 0.01 | 10.0 | 16 | 0.1 | 4 | 0.05 | 10.0 | 4 | 0.1 | 4 | 0.05 | 10.0 | 1 | 0.1 |
| med-replay | 2 | 0.05 | 10.0 | 1 | 0.1 | 4 | 0.05 | 10.0 | 8 | 0.1 | 4 | 0.05 | 5.0 | 16 | 0.1 |
| med-expert | 2 | 0.01 | 1.0 | 16 | 0.5 | 2 | 0.05 | 5.0 | 8 | 0.1 | 4 | 0.05 | 5.0 | 16 | 0.1 |
| full-replay | 4 | 0.05 | 10.0 | 4 | 0.1 | 2 | 0.05 | 10.0 | 8 | 0.1 | 2 | 0.05 | 10.0 | 1 | 0.1 |

noise $\sigma \in \{0.01, 0.05\}$, the inverse temperature parameter $\kappa \in \{0.1, 0.5, 1.0, 5.0, 10.0\}$, the number of latent samples $\bar{n} \in \{1, 4, 8, 16\}$, and the value uncertainty penalty $p \in \{0.1, 0.5, 1.0\}$. Our findings indicate that $\kappa$ and $\bar{n}$ are the most influential hyperparameters, while the others have a comparatively minor effect on performance. For LOOP, we conducted a similar grid search over the same hyperparameters, excluding $\bar{n}$, which is specific to RefPlan.

In addition, we used Bayesian optimization (BayesOpt, Snoek et al. (2012)), implemented in W&B (Biewald, 2020), to explore the challenge of identifying optimal hyperparameters for RefPlan. Figure 7 compares the number of iterations required for BayesOpt to achieve or surpass the performance of the best hyperparameter configuration found via grid search in each environment. Specifically, we used CQL as the prior policy and the MR dataset from three environments. Notably, BayesOpt required fewer than 20 iterations to exceed the performance reported in Table 2.

Table 14: Per-epoch runtimes for VAE pretraining on the ME dataset.

| | Hopper | | HalfCheetah | | Walker2d | |
|---|---|---|---|---|---|---|
| | $q_\phi$ | $\hat{p}_\psi$ | $q_\phi$ | $\hat{p}_\psi$ | $q_\phi$ | $\hat{p}_\psi$ |
| | 55.3s | 39.8s | 53.2s | 40.7s | 54.6s | 40.7s |

Table 15: Runtime per environment step for RefPlan during evaluation in the HalfCheetah environment.

| $H$ \ $\bar{n}$ | 1 | 2 | 3 | 4 |
|---|---|---|---|---|
| 2 | $7.9 \times 10^{-3}$s | $8.7 \times 10^{-3}$s | $9.3 \times 10^{-3}$s | $1.0 \times 10^{-2}$s |
| 4 | $1.5 \times 10^{-2}$s | $1.6 \times 10^{-2}$s | $1.8 \times 10^{-2}$s | $1.9 \times 10^{-2}$s |

### D.3. Computational Costs of RefPlan

In this section, we provide a detailed discussion of the computational costs associated with deploying RefPlan. As outlined in Appendix C.3, RefPlan requires the following pretrained components: a prior policy $\pi_\mathrm{p}$, an encoder $q_\phi$, and a decoder $\hat{p}_\psi$. Since the prior policy is trained using standard offline policy learning algorithms (e.g., CQL, EDAC, MOPO, COMBO, and MAPLE), which are not our contributions, we focus on reporting the computational costs associated with training the VAE model and executing the planning stage. All experiments were conducted on a single machine equipped with an RTX 3090 GPU.

**VAE Pretraining** Table 14 presents the per-epoch runtimes for VAE pretraining in the three environments. The reported runtimes correspond to datasets with 2M transition samples, the largest dataset size used in our experiments. Both the VAE pretraining and decoder fine-tuning phases were executed for up to 200 epochs or until the validation loss ceased to improve for 5 consecutive epochs, whichever occurred first.

**Test-Time Planning** At test time, planning with RefPlan involves selecting hyperparameters as detailed in Appendix D.2. Among these, the planning horizon $H$ and the number of latent samples $\bar{n}$ influence runtime. Specifically, the computational cost scales linearly with $H$, which is an inherent property of planning algorithms. However, the cost increases sub-linearly with $\bar{n}$, as shown in Table 15. For example, with $H = 4$ and $\bar{n} = 4$, the agent achieves approximately 53 environment steps per second. We hypothesize that further optimization of PyTorch tensor operations to fully exploit GPU parallelism could yield even better computational performance, particularly with respect to $\bar{n}$.

