# OpenReview forum: "Reflect-then-Plan: Offline Model-Based Planning through a Doubly Bayesian Lens"
_ICML.cc/2025/Conference — ICML 2025 poster_

### Official Review · Reviewer_BPa5 · 2025-02-17

**Overall Recommendation:** 4

**Summary:**

This paper proposes learning an approximate Bayesian model for offline RL and use planning guided by an offline RL learned policy prior for action selection. Standard practices of ensemble architecture and variance penalty are used for planning. Experiment shows improved performance over offline RL only policy and comparable or better performance over LOOP which is a similar method but does not use a VAE based model.

**Claims And Evidence:**

Yes, the experiments demonstrate improved performance across different environments and offline RL algorithms.

**Essential References Not Discussed:**

I am not aware of missed references.

**Experimental Designs Or Analyses:**

The experiment designs make sense. The chosen tasks and environments largely follow standard practices.

**Methods And Evaluation Criteria:**

It's not super clear to me the exactly problem the authors are trying to tackle, mostly the two phases the authors are referring to. It's not offline-to-online RL, or is it?

**On data sampling and evaluation**

As far as I understand:
* In phase 1: train policy prior use any offline RL algo, train encoder and decoder using the VAE objective
* In phase 2: fine-tune decoder on new data, with encoder frozen.

Section 5.1 mentions the new data are states from R. It is using R to set the initial state of the agent and the agent interacts with the environment from there on?

**On planning objective**
* The planning objective used by the authors is more of a posterior sampling approach than a Bayes adaptive approach which would require the agent to plan by rolling out $m_{t:t+h}$? The authors mentioned on line 143 that the evaluation target is single episode performance. Why not use the Bayes adaptive objective to actively reduce uncertainty and thus enhance performance in a single episode?
* The single episode objective also seems to be a bit conflicting with fine-tuning the decoder, where fine-tuning is usually done between episodes for model based RL.

**Other Comments Or Suggestions:**

NA

**Other Strengths And Weaknesses:**

NA

**Questions For Authors:**

* On planner implementation: most MPPI style planner can optimize action sequences for multiple iterations. Algorithm 2 shows that you only use a single iteration. Is that correct?

**Relation To Broader Scientific Literature:**

The paper sits within the broad offline RL literature. It takes a Bayesian perspective on offline RL which has been considered before by e.g.,  [Chen et al.](https://proceedings.neurips.cc/paper/2021/hash/470e7a4f017a5476afb7eeb3f8b96f9b-Abstract.html). Introducing online planning in this context seems to be the main novelty.

**Theoretical Claims:**

The paper is largely empirical. The theoretical motivations from the Bayesian perspective make sense.

---

> ### Author Rebuttal · Authors · 2025-04-01
>
> We thank Reviewer BPa5 for dedicating time to review our paper and for the feedback. We are glad you found the experiments demonstrate improved performance and the theoretical motivations sensible, and we appreciate the opportunity to clarify aspects of our problem setting and methodology that seemed unclear.
> 1. __Problem setting (offline vs. offline-to-online):__ Thank you for this clarifying question. Our setting is distinct from typical _"offline-to-online" RL (e.g., [1]) which often involves substantial online training iterations_. We focus on the scenario where an agent is trained entirely offline and then deployed online for evaluation, using planning to adapt its behavior based on the offline-learned model and uncertainty estimate. **No further training or model updates occur during this online deployment phase.** This "offline pre-train, deploy with planning" setting is important for applications where online interaction for training is limited, costly, or unsafe, yet adapting behavior based on learned uncertainties during deployment is crucial for robustness.
>
> 2. __Training phases (phase 1 / phase 2):__ To clarify the phases mentioned: both phase 1 (training the policy prior $\pi_{\mathrm{p}}$ and the VAE encoder/decoder) and phase 2 (fine-tuning the decoder) are performed entirely offline using the static dataset before any online interaction. The subsequent online phase is purely for evaluation without any training updates.
>
> 3. __Section 5.1 initialization:__ Your understanding is correct. For evaluation, the agent is initialized in a state sampled from the random (R) D4RL dataset and then interacts with the environment online.
>
> 4. __Planning objective (vs. Bayes-adaptive):__ RefPlan performs planning at each step using the learned model ensemble. And regarding the objective, as also clarified for Reviewer CFgz, we view the problem through the lens of an epistemic POMDP, which is an instance of a BAMDP. Therefore, indeed the underlying objective is Bayes-optimal behavior. RefPlan approximates this by sampling multiple latent model hypotheses (m) from the VAE's posterior approximation and planning within each sampled MDP using the prior-guided planner. The posterior over the MDP is continuously updated through the encoder during this evaluation time. By effectively marginalizing plans across these model samples, RefPlan seeks actions robust to the epistemic uncertainty captured by the VAE, thereby aiming to maximize performance in the evaluation episode, implicitly accounting for model uncertainty.
>
> 5. __Decoder fine-tuning.__ To clarify further based on your comment, the decoder fine-tuning mentioned (phase 2) occurs offline during pre-training, not during the online test/evaluation episode.
>
> 6. __Planner implementation (iterations):__ While Algorithm 2 presents a simplified view, our planner implementation does support multiple optimization iterations, similar to standard MPPI-style planners (e.g., resampling trajectories based on importance weights). We kept the pseudocode concise for clarity but will release our code upon acceptance to provide full implementation details. However, our earlier experiments showed no significant performance improvements from multiple sampling iterations, leading us to not pursue this approach.
>
> [1] Lee et al. (2022). “Offline-to-Online Reinforcement Learning via Balanced Replay and Pessimistic Q-Ensemble”

---

> > ### Comment · Reviewer_BPa5 · 2025-04-03
> >
> > Thank the authors for the clarification. I think the problem setting is very clear now.
> >
> > A few additional comments:
> > * On decoder fine tuning: what you mean is you first do joint training of encoder and decoder, then you freeze encoder and train decoder for some additional number of steps, all on the same offline data, is this correct? The word fine tuning is somewhat unusual. I consider this as an implementation detail. Also, could you explain why you need to do this as opposed to just stopping at joint encoder decoder training?
> > * I agree with reviewer CFgz that the work would be more elegant if uncertainty penalty was not needed for online planning. An ablation would be helpful.
> > * I still think calling the method Bayes adaptive is a bit misleading, given the planner does not plan to gather information. I think labeling this as epistemic POMDP is appropriate.

---

> > > ### Author Response · Authors · 2025-04-03
> > >
> > > Thank you for the further engagement and for confirming the clarity of the problem setting. We appreciate the chance to address your additional comments:
> > > 1. __Decoder training steps:__ Your understanding is correct. We first train the encoder and decoder jointly, then freeze the encoder and continue training the decoder for a few extra epochs, all using the same offline dataset $\mathcal{D}$. We agree that calling the latter part "fine-tuning" might be confusing given the same data source and will revise the phrasing for clarity. Thank you for pointing this out.
> > > 2. __Rationale for additional decoder training:__ Our rationale for these extra steps relates to a key difference from VariBAD: RefPlan directly uses the learned decoder model for planning at test time. Therefore, maximizing its predictive accuracy is crucial to minimize the impact of model error on planning. Empirically, these extra decoder steps reduced validation loss (on held-out offline data), suggesting better predictive accuracy for planning. Since the decoder plays a key role during planning in RefPlan, we believed it was relevant to note this aspect in the main text.
> > > 3. __Uncertainty penalty ablation:__ Thank you for echoing Reviewer CFgz's point on the uncertainty penalty. We agree that demonstrating performance without relying on conservatism is important. As requested, we performed this ablation for our initial response to Reviewer CFgz. For self-containedness, we include the results also here for your reference:
> > >
> > > _[Table comparing RefPlan with and without the penalty, identical to the one provided in our response to Reviewer CFgz]_
> > >
> > > | Env | Config | RefPlan| RefPlan w/o penalty |
> > > |----|----|----|----|
> > > | Hopper | MR | 98.1 ± 0.5 | 98.26 ± 0.5 |
> > > | Hopper | FR | 107.6 ± 0.5 | 107.71 ± 0.5 |
> > > | Walker2d | MR | 93.6 ± 0.3 | 93.71 ± 0.2 |
> > > | Walker2d | FR | 101.6 ± 1.1 | 100.35 ± 1.4 |
> > > | Halfcheetah | MR| 54.1 ± 0.6 | 54.34 ± 0.3 |
> > > | Halfcheetah | FR | 86.7 ± 0.7 | 87.42 ± 0.9 |
> > >
> > >  As the results show, strong performance is maintained without the penalty under the considered tasks, indicating the primary gains stem from effectively utilizing epistemic uncertainty via Bayes-adaptive planning, rather than the penalty term.
> > >
> > >
> > > 4. __Bayes-adaptiveness:__ We appreciate you raising the insightful point about the connection to posterior sampling and the nature of planning in RefPlan; this is a valuable perspective that we may have overlooked initially. Indeed, our approach can be viewed as approximately optimizing the Bayes-adaptive objective via _posterior sampling_: at each step, we sample MDP hypotheses (m) based on the current belief (from the encoder) and plan within them.
> > >
> > > While explicitly planning to gather information (i.e., updating the latent belief during the planning rollout) is another way to approach BAMDPs, our preliminary experiments suggested this could be detrimental. _Actions deemed informative within an imperfect learned model might not translate to effective information gathering actions in the true environment due to model errors._
> > >
> > > Consequently, we adopted the posterior sampling approach which still leverages the belief over MDPs to make robust decisions under uncertainty, but avoids potentially misleading information-seeking based on flawed model rollouts. We appreciate you highlighting this connection, and we will add discussion clarifying the relationship between our method and posterior sampling approaches for BAMDPs in the revision.
> > >
> > >
> > > Finally, we thank you again for the valuable discussion and insightful comments, which have helped improve the paper. We hope our clarifications are helpful and lead you to positively reassess your evaluation.

---

### Official Review · Reviewer_GXYS · 2025-02-22

**Overall Recommendation:** 3

**Summary:**

The authors introduce RefPlan, a doubly Bayesian method for offline model-based RL.
RefPlan combines two existing methods (1) the probabilistic control-as-inference formulation of MB planning (using a policy prior) with (2) the variational representation of epistemic uncertainty.
At inference time, RefPlan marginalizes over the latent variable representing the environment, to consider several possible MDPs and improve inference-time planning.
The authors show that on D4RL, RefPlan is robust to OOD states, to environment changes and performs well with limited data.

**Claims And Evidence:**

The experiments ask interesting questions and study whether Replan
 - mitigates performance drop due to OOD states (5.1)
 - enhances performance and outperforms other methods (conservative policies) (5.2)
 - performs well with subsampled datasets (5.3)
 - is robust to changing dynamics (5.4)

However, I have two comments about the experiments:

(1) I wish the authors had used more than 3 seeds, which is very little and insufficient to claim statistical significance. Also, their error bars are missing in Table 1, Table 2 and Figure 3.

(2) In the current setup, RefPlan increases the inference budget by a factor $\bar{n}$ (the number of times the latent variable m is sampled). Consequently, the authors should show that the performance gains reported do not simply come from this increase, but from combining the strengths of their two Bayesian frameworks.
See my questions (2), (3), (4) below.

**Essential References Not Discussed:**

Since the authors discuss MB offline policy learning (also known as background planning) they should cite the original Dyna work
  - Sutton, Richard S. "Integrated architectures for learning, planning, and reacting based on approximating dynamic programming." Machine learning proceedings 1990. Morgan Kaufmann, 1990. 216-224.
and the Dreamer line of work:
   - Hafner, Danijar, et al. "Mastering diverse domains through world models." arXiv preprint arXiv:2301.04104 (2023).

In the context of planning as inference, the authors should cite the sequential Monte Carlo work, which estimates the probability of following an optimal trajectory from the current timestep and add a resampling step.
  - Piché, Alexandre, et al. "Probabilistic planning with sequential monte carlo methods." International Conference on Learning Representations. 2018.

**Experimental Designs Or Analyses:**

See "Claims And Evidence"

Also, the authors only report normalized scores (100 being online SAC). They should report the SAC scores, so that future work can compare to them.

**Methods And Evaluation Criteria:**

The proposed framework which combines (1) a probabilistic inference for MP planning, using a learned policy as prior and (2) a latent variable for modeling the underlying MDP makes sense.

**Other Comments Or Suggestions:**

(1) Equation (1): $\hat{r}_{\psi}$ has not been defined. I assume it is a learned reward model.

(2) The optimism bias that occurs with the RL as inference framework should probably be mentioned.

**Other Strengths And Weaknesses:**

I enjoyed reading the paper, which was well written and easy to follow.

However, I think that the paper would benefit from highlighting more the authors' contribution. Here is one option:

 - As far as I understand, Sections 4.1 and 4.2 are summarizing the MBOP and VariBAD work, and the main novelty of the paper is 4.3. I find the current 4.1 and 4.2 to be misleading: I think the authors should consider moving these section to the preliminaries Section 3. They should also consider compressing Section 2 and 3, so that their contribution starts before the 6th page---which is currently the case.

 - In this new organization, Section 4 would be focused on RefPlan. The authors may also consider moving Algorithm 2 to Section 4.

 - The authors should keep more space for their experiments. For instance I am not sure that the results for applying RefPlan to MAPLE and COMBO should be in the appendices (there could be a table in the main text).

**Questions For Authors:**

(1) Is there any technical novelty in Sections 4.1 and 4.2? As far as I understand, 4.1 is similar to MBOP which also uses a BP policy as action prior, and 4.2 is summarizing VariBAD?

(2) What is the value of $\bar{n}$ and $\bar{N}$ used in the experiments? How does it compare to LOOP?

(3) Could the authors show that introducing the latent variable $m$ is useful? One option would be to compare: RefPlan vs. planning as inference without epistemic uncertainty (Equation 7) using using $\bar{n} \times \bar{N}$ samples

(4) Similarly, could the authors compare RefPlan to others method for $\textbf{the same}$ inference budget? Two options could do that could be:
  - using $\bar{n} \times \bar{N}$ samples for LOOP (same as RePlan)
  - only replanning the sampled every $\bar{n}$ step when using RefPlan

I am willing to increase my score if the authors can provide a stronger evidence that their gains come from their proposed RefPlan and not from simply increasing the inference budget of existing methods.

**Relation To Broader Scientific Literature:**

Increasing run-time compute for improving the performance of pre-trained agents is a critical problem for developing general agents. The paper proposes an interesting approach in this direction.

**Theoretical Claims:**

The paper does not contain theoretical claims.

---

> ### Author Rebuttal · Authors · 2025-04-01
>
> Thank you for the detailed, constructive review. We appreciate the opportunity to address your feedback, particularly on statistical significance and budget comparisons.
> 1. Statistical significance: While runs use 3 seeds, Figure 5 uses RLiable [1] for robust aggregate analysis (Appx B.1). Figure 5 demonstrates that RefPlan consistently outperforms LOOP across aggregate metrics with non-overlapping confidence intervals, indicating statistically meaningful improvements. We will add error bars to Figure 3 on revision.
> 2. Reporting raw scores: Normalized scores are from the D4RL codebase; we will clarify this on revision.
> 3. Essential references: We will add discussion on Dyna/Dreamer. Re: SMC, we cited Piche et al. (lines 114, 166) but will expand discussion on its relation to RefPlan in revision.
> 4. Paper structure & highlighting contributions: Thank you for the thoughtful suggestions on restructuring. We agree that improving the flow to present novel aspects earlier could strengthen the paper. While major restructuring is challenging, we commit to revising the presentation for the camera-ready version to improve clarity and better foreground RefPlan's core contributions.
> 5. Answers to specific questions:
>
>      1) Novelty in Sec 4.1 & 4.2: While algorithmically related to MBOP/VariBAD, our primary contribution is the conceptual framing and synthesis. Deriving MB planning via control-as-inference justifies the prior and enables the Bayesian uncertainty treatment (Sec 4.3). The novelty lies in integrating these for offline RL in RefPlan. We will clarify this. We will refine the writing to make this relationship clearer.
>      2) $\bar{n}$ and $\bar{N}$ values: For the main results (RQ2), we used $\bar{n}=16$ and the same $\bar{N}$ as LOOP. Full hyperparameter details are in Appendix D.2, Table 7.
>     3) Usefulness of latent variable: Fig 8 shows benefits up to $\bar{n}=16$. New results with $\bar{n}=32,64$ (in our response to Reviewer CFgz, Tables 2-3), show further gains, particularly under high uncertainty due to limited-data (RQ3), suggesting the value of marginalizing over $m$.
>     4) Comparison with same inference budget: This is a crucial point. We conducted new experiments to directly address this:
> * LOOP with increased budget: We compared RefPlan against LOOP using 16x its original sampling budget (matching RefPlan's max $\bar{n}$). Table 4 shows RefPlan generally maintains an advantage, though LOOP benefits from compute (esp. w/o penalty in HalfCheetah-FR).
>
> Table 4: RefPlan vs. LOOP-16x budget, RQ2
> |Env|Config|LOOP|LOOP w/o penalty|RefPlan|
> |---|---|---|---|---|
> |Hopper|MR|97.8 ± 1.1|97.8 ± 0.8|98.1 ± 0.5|
> | |FR|107.5 ± 0.6|107.5 ± 0.6|107.6 ± 0.5|
> |Walker2d|MR|83.2 ± 8.8|78.4 ± 7.1| 93.6 ± 1.1 |
> | | FR | 99.9 ± 1.5 | 99.9 ± 1.5 | 101.3 ± 0.3 |
> | Halfcheetah | MR | 53.2 ± 0.1 | 55.1 ± 0.4 | 54.1 ± 0.6 |
> | | FR | 83.1 ± 0.8 | 89.4 ± 0.7 | 86.7 ± 0.7 |
> * LOOP with increased budget (limited data): We repeated this comparison in the RQ3 setting (Table 5). Here, RefPlan's advantage over the high-budget LOOP was often more pronounced, especially with smaller datasets, suggesting RefPlan better handles the higher epistemic uncertainty.
>
> Table 5: RefPlan vs. LOOP-16x budget, RQ3
>
> Hopper FR:
> | Data Size | LOOP| RefPlan|
> |----|----|----|
> | 50k | 101.7 ± 4.9| 99.5 ± 12.9|
> | 100k| 106.9 ± 0.3| 107.0 ± 0.7|
> | 250k | 107.2 ± 0.3| 106.8 ± 0.6|
> | 500k | 104.5 ± 3.6| **107.7 ± 0.4**|
> Walker2d FR:
> | Data Size | LOOP| RefPlan|
> |----|----|----|
> | 50k| 70.6 ± 16.4 | **82.1 ± 9.5**|
> | 100k| 95.9 ± 1.4 | 96.4 ± 2.1|
> | 250k| 96.1 ± 0.8 | 96.8 ± 1.2|
> | 500k| 98.6 ± 0.9 | **100.8 ± 0.8**|
> Halfcheetah FR:
> | Data Size | LOOP | RefPlan |
> |----|----|----|
> | 50k| 63.8 ± 0.2| **68.5 ± 1.6** |
> | 100k| 71.4 ± 0.4| **75.7 ± 1.0**|
> | 250k| 77.9 ± 0.3 | **81.8 ± 0.3**|
> | 500k| 82.4 ± 0.7 | **83.9 ± 1.4**|
> * RefPlan with reduced budget: We matched RefPlan's budget to LOOP's by increasing its replanning interval. Table 6 shows RefPlan still performed competitively, often better than LOOP (except HalfCheetah-FR).
>
> Table 6:
> |Env|Config|LOOP|RefPlan-ReducedFreq|
> |----|----|----|----|
> | Hopper| MR| 97.5 ± 0.5| **98.7 ± 0.6**|
> |  | FR  | 106.2 ± 0.7| **107.8 ± 0.5**|
> | Walker2d| MR| 81.9 ± 3.0| **89.3 ± 2.3**|
> | | FR  | 99.4 ± 0.3| **100.1 ± 0.5** |
> | Halfcheetah | MR| 52.1 ± 0.0| 52.1 ± 0.2 |
> | | FR | **81.8 ± 1.0**| 79.4 ± 0.7|
> These results suggest that RefPlan's performance gains are not merely due to increased computation but stem from the way it handles epistemic uncertainty via the doubly Bayesian approach, leading to more effective planning.
>
> We hope these responses, new experiments directly addressing the budget comparison, and our commitments to revision clarify the contributions and robustness of RefPlan. We are grateful for your constructive feedback and hope we have provided the evidence needed to reconsider the initial evaluation.
>
> [1] Agarwal et al. (2021). “Deep Reinforcement Learning at the Edge of the Statistical Precipice”

---

> > ### Comment · Reviewer_GXYS · 2025-04-03
> >
> > I thank the authors for carefully addressing my comments and running these additional experiments. They indeed show that RefPlan can lead to some performance gain w.r.t. LOOP, when using the same inference budget. Part of these should be included in the revised paper.
> >
> > Although the camera ready version of the paper will require significant restructuring + will need to include many of the additional experiments run in the rebuttals, I am happy to increase my score.

---

> > > ### Author Response · Authors · 2025-04-05
> > >
> > > Thank you for carefully considering our response and for the positive reassessment. We acknowledge the need for significant revisions for the camera-ready version, including restructuring and incorporating the new experimental results. We are fully committed to making these improvements should the paper be accepted.

---

### Official Review · Reviewer_CFgz · 2025-03-10

**Overall Recommendation:** 3

**Summary:**

This paper combines ideas from _adaptive_ and _online planning_ Offline RL to achieve a conceptually nice framework and reasonable performance improvements. They are able to improve upon (a) epistemically adaptive methods with no online computation, and (b) online computation methods with no explicit epistemic adaptability.

**Claims And Evidence:**

The claims made are broadly sensible and supported, other than:

1. The claim that "existing methods rely on fixed conservative policies" (abstract lines 15-17) is an oversimplification and is contradicted by later discussion of several adaptive ORL methods. The authors should clarify this point so the claims are not overstated.
1. I don't agree with the way Epistemic POMDPs are compared to BAMDPs (lines 126-155, 185-191).
    - There is no real conceptual difference between the Epistemic POMDP and BAMDP formulation, or the methods used to solve them. The difference is purely perspective/assumption-based, where epistemic POMDPs were pitched with the offline->online deployment setting in mind. The description in the text ("Epistemic POMDPs prioritize performance during a single evaluation episode") makes it seem as if there is a fundamental difference in their representative capabilities, which is not the case. This is even said later in the paper -- "an epistemic POMDP is an instance of a BAMDP" (lines 185-186).
    - Along the same lines: line 202-203 says "we can leverage the BAMDP reformulation of epistemic POMDPs". I don't think there's any reformulation needed.
    - I don't think it’s accurate to say a “BAMDP can be reformulated as a belief MDP” (line 130); a BAMDP is itself a special case of a belief MDP [Zintgraf et al. VariBAD: A very good method for bayes-adaptive deep rl via meta-learning, ICLR 2020.]
    - Ultimately I think this confusion arises from the original Epistemic POMDP paper [Ghosh et al. 2021], where I do wish they'd originally posed the problem as a BAMDP. But there's no need to perpetuate that confusion here.

**Essential References Not Discussed:**

I did not know of or find any related work that was essential to be cited in this paper.

**Experimental Designs Or Analyses:**

The basic experimental approach seems sound.

### Ablations

I would have liked to see ablation studies or some kind of qualitative comparison to justify the design choices and components of the method:

1. "Additionally, following Sikchi et al. (2021), we apply an uncertainty penalty based on the variance of the returns predicted by the learned model ensemble" (316-320), would be nice to see an ablation of this. It would be more elegant if all behaviour was Bayes-adaptive rather than some conservatism.
1. Ablating prior policy with random policy instead

### Statistical significance

Although 3 seeds is not very many for the experiments, experiments are repeated several times over different prior policy methods, so the results in Figure 5 are based on a reasonable number of samples. It would be nicer if this plot was in the main body of the paper.

### Experiment improvements

- I would like to see more latent sample values being tried (Figure 8), rather than 16 being the maximum number.
    - The "normalised score" hasn't obviously plateaued in the range of latent sample values shown in Figure 8.
    - MCTS-based Bayes-adaptive planning methods such as BAMCP [Guez et al., "Scalable and efficient Bayes-adaptive reinforcement learning based on Monte-Carlo tree search". JAIR 2013.] effectively sample a new MDP (equivalent to latent sample) for each MCTS trial, which would be >> 16 samples.
    - Given everything seems to run relatively fast (Table 13) I don't see why this couldn't be done with 32 or 64 samples for example.
- It would be nice if stochastic MDPs were evaluated, as well as the D4RL deterministic continuous control ones. Evaluating only on deterministic environments is common for offline RL papers however.

**Methods And Evaluation Criteria:**

The chosen combination of methods is sensible, and the evaluation domains and baselines are standard for offline RL.

**Other Comments Or Suggestions:**

## Minor / clarity

1. Figures 3/6/7 seem like they could be combined somehow, with a more information-dense presentation than bar charts. It could also do with more discussion in the text, as the performance improvement over LOOP seems weakest in this experiment setting.
1. I think clarity would be aided by always showing subscripts on $\mathcal{O}$ in (2) and (3)
1. The list of evaluation configurations (lines 351-355) could make it clearer that it is the behaviour policy that is changing between configurations. It would be helpful to have a brief description of how these configurations vary, or a pointer to somewhere where the variants are more fully described, especially for a reader without familiarity with D4RL.


### Typos

- Line 43: “have severe implications” should be “has severe implications”
- Line 164: “model of the environment refers to the transition and reward functions”. The difference between the ground-truth T and r, and the estimated models of them, should be made clearer here.
- Line 973 “moel” -> “model”

**Other Strengths And Weaknesses:**

1. The flow and clarity of the paper is good. Given that this paper combines ideas/algorithms from several fields, the authors achieved the task of succinctly covering the important concepts and how they are combined into their method.

**Questions For Authors:**

1. Did you carry out ablations or comparisons to justify your design choices discussed in my ablation comments? As a combination-of-methods paper I think it is important to discriminate between the contributions of the individual components, and ablation studies or more discussion would help with this.
1. Do you agree with my comments on the Epistemic POMDP/BAMDP confusion? I could be convinced otherwise if you have a good argument.

**Relation To Broader Scientific Literature:**

The work is a combination of ideas from two main areas in offline RL: adaptive behaviour and online planning. The contribution of this paper is to combine these ideas in a way that is conceptually nice.

**Theoretical Claims:**

I checked the derivations in the text, which are relatively simple (the result of minor changes to existing methods). These derivations seem correct to me.

---

> ### Author Rebuttal · Authors · 2025-04-01
>
> We sincerely thank Reviewer CFgz for taking the time to thoroughly analyze our paper and provide constructive and insightful feedback.
> We address the main points raised below:
> 1. Epistemic POMDP vs. BAMDP: Thank you for this important point. We agree there's no fundamental conceptual difference and "reformulation" was imprecise. Our use of the Epistemic POMDP perspective aimed to highlight our specific offline pre-training -> single online evaluation setting, following [1]. As noted (lines 185-186), an Epistemic POMDP is an instance of a BAMDP; the distinction is mainly perspective. We apologize for the confusion and commit to revising the text for the camera-ready version to clarify this relationship and correct related statements (e.g., line 130).
> 2. Ablation on uncertainty penalty: This is an excellent suggestion. To disentangle the benefits of Bayes-adaptivity from conservatism, we conducted new ablation experiments removing the uncertainty penalty, focusing on the CQL prior with medium-replay (MR) and full-replay (FR) datasets.
>
> [Table 1]
> | Env | Config | RefPlan| RefPlan w/o penalty |
> |----|----|----|----|
> | Hopper | MR | 98.1 ± 0.5 | 98.26 ± 0.5 |
> | Hopper | FR | 107.6 ± 0.5 | 107.71 ± 0.5 |
> | Walker2d | MR | 93.6 ± 0.3 | 93.71 ± 0.2 |
> | Walker2d | FR | 101.6 ± 1.1 | 100.35 ± 1.4 |
> | Halfcheetah | MR| 54.1 ± 0.6 | 54.34 ± 0.3 |
> | Halfcheetah | FR | 86.7 ± 0.7 | 87.42 ± 0.9 |
>
> The table shows strong performance is maintained without the penalty, indicating gains primarily stem from Bayes-adaptive planning utilizing epistemic uncertainty, not conservatism.
>
> 3. Ablation with random prior policy: We appreciate the suggestion. Early experiments showed random priors performed very poorly due to distribution shift. Given that even stronger priors like BC policies (e.g., in MBOP) were outperformed by LOOP, we decided a full sweep with a random prior would be less informative than other comparisons.
> 4. Number of latent samples: Thank you for this keen observation regarding Figure 8 and the potential benefits of using more latent samples. We ran new RQ2 experiments with 32 and 64 latent samples (CQL prior, MR/FR datasets). Table 2 compares these with $\bar{n}=16$.
>
> [Table 2]
> |Env|Config|16|32|64|
> |---|---|---|---|---|
> |Hopper|MR|98.3±0.5|96.5±0.3|98.0±0.3|
> | |FR|107.6±0.5|107.6±0.5|107.6±0.6|
> |Walker2d|MR|92.8±0.7|88.1±2.0|87.1±3.8|
> | |FR|100.1±0.8|100.3±0.9|100.3±1.3|
> |Halfcheetah|MR|54.3±0.3|54.1±0.3|54.4±0.1|
> | |FR|87.4±0.9|87.1±0.9|87.2±0.7|
>
> In this standard setting, increasing $\bar{n}$ beyond 16 yielded no significant gains, possibly due to lower epistemic uncertainty. We hypothesized the effect would be clearer with higher uncertainty. To test this, we ran additional RQ3 experiments using models trained on subsampled datasets.
>
> [Table 3]
> |Env|Data Size|16|32|64|
> |---|---|---|---|---|
> |Hopper|50k|99.5±12.9|102.7±4.0|**106.2±3.3**|
> |Hopper|100k|107.0±0.7|106.9±0.4|106.9±0.4|
> |Hopper|250k|106.8±0.6|103.2±3.3|106.7±0.2|
> |Hopper|500k|107.7±0.4|107.7±0.5|107.5±0.6|
> |Walker2d|50k|82.1±9.5|85.3±10.9|**93.9±1.0**|
> |Walker2d|100k|96.4±2.1|89.5±6.2|84.4±5.9|
> |Walker2d|250k|96.8±1.2|96.8±0.6|96.4±0.6|
> |Walker2d|500k|100.8±0.8|99.2±0.3|99.9±0.4|
> |Halfcheetah|50k|68.5±1.6|69.1±1.0|**69.3±1.0**|
> |Halfcheetah|100k|75.7±1.0|76.2±0.3|76.1±0.6|
> |Halfcheetah|250k|81.8±0.3|82.0±0.2|82.1±0.5|
> |Halfcheetah|500k|83.9±1.4|83.8±1.1|84.0±1.0|
> As shown in the table, with higher uncertainty (limited data), increasing $\bar{n}$ to 64 does indeed lead to more noticeable performance gains, particularly with the 50k dataset. This confirms the value of more samples in these settings.
>
> 5. Stochastic environments: We agree that evaluating on stochastic environments would be a valuable extension. We acknowledge this limitation and plan to explore this direction in future work.
>
> 6. Minor points & typos: Thank you for pointing out the areas for clarification and the typos. We will address all these points in the camera-ready version.
>
> We hope these clarifications, new experimental results, and our commitment to revise the paper address your concerns. We are grateful for the constructive feedback, which has immensely helped us strengthen the paper. We believe the results, particularly the new ablations and latent sample experiments, further validate the effectiveness of our approach.
>
> [1] Ghosh et al. (2021) “Why Generalization in RL is Difficult: Epistemic POMDPs and Implicit Partial Observability”

---

> > ### Comment · Reviewer_CFgz · 2025-04-04
> >
> > Thank you for the clarification and additional experiment results. I will leave my recommendation unchanged.
> > Table 3 is interesting and answers my question on behaviour with respect to the number of latent samples. I think the paper or appendices should include this discussion on the link between small training data and the benefit of using more latent samples. The small data / high model uncertainty regime is key to demonstrating the benefits of Bayes adaptive methods.

---

> > > ### Author Response · Authors · 2025-04-05
> > >
> > > Thank you for acknowledging our response and for the additional feedback. We are glad the results on the number of latent samples were informative. We agree that discussing the link between limited data/high uncertainty and the benefit of more latent samples is important. We will certainly incorporate this discussion into the camera-ready version, either in the main text or appendices, if the paper is accepted.

---

### Decision · Program_Chairs · 2025-05-01

**Decision:**

Accept (poster)

**Comment:**

The RefPlan method is a useful approach for offline RL. It uses both online planning (guided by a policy prior) and a way to handle uncertainty about the environment (using latent MDPs). Tests showed it got better results and was reliable on hard problems. Reviewers initially had some questions about the experiments and explanations. However, the authors addressed some of the experiment problems in their response, which helped support their results. Since the approach is new and works well, the overall view is positive and the paper should be accepted.